

# Possible superconductivity from incoherent carriers in overdoped cuprates

M. Čulo[1,2⋆†], C. Duffy[1† ‡], J. Ayres[1,3], M. Berben[1], Y.-T. Hsu[1],
R. D. H. Hinlopen[3], B. Bernáth[1] and N. E. Hussey[1,3∘]

**1** High Field Magnet Laboratory (HFML-EMFL) and Institute for Molecules and Materials,
Radboud University, Toernooiveld 7, 6525 ED Nijmegen, Netherlands
**2** Permanent address: Institut za fiziku, P.O. Box 304, HR-10001 Zagreb, Croatia
**3** H. H. Wills Physics Laboratory, University of Bristol,
Tyndall Avenue, Bristol BS8 1TL, United Kingdom

⋆ matija.culo@ru.nl, ‡ caitlin.duffy@ru.nl, ∘ n.e.hussey@bristol.ac.uk

## Abstract

There is now compelling evidence that the normal state of superconducting overdoped cuprates is a strange metal comprising two distinct charge sectors, one governed by coherent quasiparticle excitations, the other seemingly incoherent and characterized by non-quasiparticle (Planckian) dissipation. The zero-temperature superfluid density $n_s(0)$ of overdoped cuprates exhibits an anomalous depletion with increased hole doping $p$, falling to zero at the edge of the superconducting dome. Over the same doping range, the effective zero-temperature Hall number $n_H(0)$ transitions from $p$ to $1 + p$. By taking into account the presence of these two charge sectors, we demonstrate that in the overdoped cuprates $Tl_2Ba_2CuO_{6+\delta}$ and $La_{2-x}Sr_xCuO_4$, the growth in $n_s(0)$ as $p$ is decreased from the overdoped side may be compensated by the loss of carriers in the coherent sector. Such a correspondence is contrary to expectations from conventional BCS theory and implies that superconductivity in overdoped cuprates emerges uniquely from the sector that exhibits incoherent transport in the normal state.

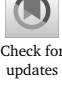

## Contents

† These authors contributed equally to this work

# 1  Introduction

As in many other unconventional superconductors, the transition temperature $T_c$ of high-$T_c$ cuprates traces out a dome in their phase diagram. In cuprates, this dome is parameterised by a range of doping $p$ (across which superconductivity appears) and a maximum $T_c$ at optimal (OP) doping. The reasons for the loss of superconductivity on either side of the dome are not yet well understood, though it is generally believed that on the underdoped (UD) side, proximity to the Mott insulating state is key [1], while on the overdoped (OD) side, superconductivity vanishes due to a diminishing pairing interaction [2,3]. The anomalously low superfluid density $n_s(0)$ found early on in OD cuprates [4,5] was then attributed to pair breaking, following standard BCS treatments for a disordered or 'dirty' $d$-wave superconductor [6].

A challenge to this viewpoint emerged in a recent study of the superfluid density in OD $La_{2-x}Sr_xCuO_4$ (LSCO) thin films [7]. There, the decrease in $n_s(0)$ on approach to the edge of the superconducting (SC) dome at $p_{SC} \sim 0.27$ was mapped out in great detail. The authors argued that while pair-breaking due to disorder can reduce $n_s(0)$, the levels of pair-breaking required within conventional BCS theory would render the $T$-dependence of the superfluid density $n_s(T)$ quadratic over a wide temperature range, in contrast to the observed ('clean') $T$-linear behavior. Its explanation, the authors concluded [7], lay outside the realms of BCS theory.

The key requisite of a BCS superconductor is a Fermi-liquid (FL) ground state with a well-defined Fermi surface (FS). In OD $Tl_2Ba_2CuO_{6+\delta}$ (Tl2201) with $T_c < 30$ K ($p > 0.27$), the observation of quantum oscillations (QO) appeared to confirm the existence of a large hole-like FS containing $1 + p$ carriers (corresponding to the full Luttinger count) [8]. As such, OD cuprates are commonly perceived to be conventional, both in their normal and superconducting states [9]. In reality, the normal state transport properties of all OD cuprates, including Tl2201, are far from conventional. This so-called 'strange metal' regime has three notable characteristics: (i) a ubiquitous non-FL ($T$-linear) component in the in-plane resistivity $\rho_{ab}(T)$ at low $T$ [10–12], whose coefficient $\alpha(0)$ scales with $T_c$ and is consistent with a scattering rate at the Planckian dissipation limit $\hbar/\tau \sim k_B T$ [12,13]; (ii) a Hall number $n_H(0)$ deduced from

the low-$T$ Hall effect that does not follow the expected 'Luttinger' $1 + p$ line but instead drops monotonically towards $p$ near OP doping [14] and (iii) a $H$-linear magnetoresistance (MR) at high field strengths [15] that exhibits $H/T$ scaling [16] and is also insensitive to both field orientation and impurity scattering rate [16] (for more details, see the introduction to appendix A).

There are several possible origins for Planckian $T$-linear resistivity, including electron-phonon scattering [17], 'hot-spot' electron-electron scattering [18] and scattering off quantum critical fluctuations [17]. Given that the Debye temperature $\Theta_D \approx 400$ K in cuprates [19], it is unfeasible to associate $T$-linear resistivity that persists down to the mK range [10, 11] to electron-phonon scattering. A Fermi surface containing hot spots – regions of high density of states, e.g. where the Fermi level $\epsilon_F$ crosses a van Hove singularity (vHs) – may result in a $T$-linear resistivity down to 0 K [18]. While the two cuprate families considered here are known to host a vHs crossing somewhere in their phase diagram, across the doping region of interest $(0.20 < p < 0.30)$, $\epsilon_F$ in LSCO is tuned away from the vHS, while in Tl2201, $\epsilon_F$ is tuned towards it [14]. The evolution of the $T$-linear coefficient with doping in both systems, however, is very similar [12, 13]. Hence, the vHs itself cannot be responsible for the $T$-linearity of $\rho(T)$ in OD cuprates. Thus we conclude that this $T$-linear resistivity is a signature of incoherence and maximal dissipation [20–22], possibly associated with quantum critical fluctuations of as yet unknown origin. On more general grounds, the condition for quasiparticle coherence is met once its decay rate $\Gamma$ becomes smaller than its excitation energy $\epsilon$ which, in a FL, is guaranteed by the relation $\Gamma \sim \epsilon^2$. In strange metals, on the other hand, quasiparticle decoherence is implicit in the linear dependence of $\Gamma$ (or $\rho$) on $T$ and $\epsilon$ at the lowest temperatures and energies, as well as in its associated Planckian timescale.

The insensitivity of the $H$-linear MR to field orientation and impurity scattering is yet another signature of incoherent transport in OD cuprates. Such insensitivity contrasts markedly with expectations from Drude or Boltzmann transport theory in which the magnitude of the MR is dictated both by the strength of the Lorentz force and by the product of the cyclotron frequency and scattering time $\omega_c \tau$, where $\tau$ includes both inelastic and elastic (i.e. impurity) scattering. Moreover, the observed $H/T$ scaling of the MR [16] implies a direct link between the non-orbital MR and Planckian dissipation. The drop in the low-$T$ Hall carrier density $n_H(0)$ with decreasing doping [14] – a drop that is larger than any residual field dependence in $R_H$ at low $T$ [14, 23] – can also be viewed as a signature of carriers with no intrinsic Lorentz-driven Hall response. Overall, these various transport anomalies reveal a consistent picture in which quasiparticle coherence is gradually suppressed as optimal doping is approached from the overdoped side.

The form of the (magneto)resistivity found in OD cuprates is similar to that observed in other correlated metals close to a QCP [24–26]. Uniquely, in OD cuprates, this $T$- and $H$-linear resistivity exists across the entire strange metal regime [16], suggesting that it is not tied to any singular QCP [27]. Similarly, the observed loss of carriers across this regime is difficult to reconcile with the absence of any pseudogap features, e.g. in the electronic specific heat [28] above $p^*$. Collectively, these features establish OD cuprates as exceptional non-Fermi-liquids harboring two distinct charge sectors; one associated with coherent quasiparticles, the other incoherent and non-quasiparticle in nature [16].

Given the presence of these two sectors, it is pertinent to pose the question: which sector is responsible for (high-temperature) superconductivity? Here we show, with a minimal set of assumptions, that with decreasing doping, the superfluid density at 0 K ($n_s(0)$) in Tl2201 and LSCO grows at the expense of the coherent carrier density ($n_{coh}$). This correspondence leads us to postulate that superconductivity within the strange metal phase of OD cuprates is not, as expected, an instability of the FL, but rather an instability of the incoherent non-FL sector.

## 2  Superfluid and coherent carrier density in Tl$_2$Ba$_2$CuO$_{6+\delta}$

The key quantities for our analysis are $n_H(0)$ and $n_s(0)$, the Hall number and superfluid density per Cu atom, respectively. For a single-layer cuprate like Tl2201 with a barrel-shaped FS [29,30] and an almost isotropic (in-plane) Fermi velocity $v_F$ or effective mass $m^*$, the former can be expressed simply as

$$n_H(0) = V_{cell}/(R_H(0)e), \tag{1}$$

where $R_H(0)$ is the as-measured Hall coefficient in the low-$T$, high-field limit (i.e., once any residual anisotropy is washed out [14]) and $V_{cell}$ is the unit cell volume. Similarly, $n_s(0)$ can be derived from the London equation:

$$n_s(0) = (m^*V_{cell})/(\mu_0 e^2 \lambda_{ab}^2(0)), \tag{2}$$

where $\lambda_{ab}(0)$ is the in-plane zero-temperature penetration depth. In an ordinary one-component Galilean invariant FL, correlation effects do not cause a renormalisation of $\lambda(0)$ and hence, one must use the bare electron mass to estimate $n_s(0)$ [31]. In heavy fermion systems [32] and iron-pnictides [33], however, $\lambda_{ab}(0)$ is found to be renormalised by the thermodynamic mass $m^*$, the breaking of Galilean invariance attributed to the presence of two independent components. Similarly, we infer here that the breaking of translational invariance in OD cuprates may also result from the existence of the two sectors, though other factors, including strong correlations [34], disorder and/or Umklapp scattering could also be playing a role.

To motivate the inclusion of the thermodynamic mass in Eq. (2) for OD cuprates, we compare in Figure 1A the fraction of carriers that condense into the superconducting state in OD Tl2201 determined via two independent routes. The closed triangles in Figure 1A represent the ratio $n_s(0)/(1+p)$, where $n_s(0)$ has been extracted from muon-spin relaxation ($\mu$SR) [4,5,35] and microwave surface impedance [6] measurements of $1/\lambda_{ab}^2(0)$, using Eq. (2) and assuming $m^* = 5.2\,m_e$. This doping-independent value for $m^*$ matches that obtained from quantum oscillation (QO) experiments on single crystals with $T_c = 10$ K and 27 K [37] as well as the normal state electronic specific heat coefficient $\gamma_N$ for 0 K $\leq T_c \leq$ 80 K [28]. The values for $(1+p)$ – the full Luttinger count – are determined from the value of $T_c$ as described in Ref. [14]. The circles in Fig. 1A represent $\Delta\gamma(0)/\gamma_N$ where $\Delta\gamma(0) = \gamma_N - \gamma(0)$ and $\gamma(0)$ is the residual electronic specific heat in the zero-temperature limit [36]. Hence, $\Delta\gamma(0)/\gamma_N$ represents the fraction of states that enter into the condensate. Note that both fractions appear to approach unity around $p = 0.19$. (See appendices A.6 and A.7 for more details on how these values were obtained.)

It is important to realise that such excellent agreement between the two quantities shown in Fig. 1A is not a trivial finding. While $\Delta\gamma(0)/\gamma_N$ is independent of $m^*$, the conversion from $1/\lambda_{ab}^2(0)$ to $n_s(0)$ in Eq. (2) relies solely on the magnitude of $m^*$, which is itself determined independently by measurements of $\gamma_N$ (as well as QO). Rather, this robust consistency between the two quantities – ostensibly across the entire OD regime of Tl2201 – affirms the need to invoke the thermodynamic mass in the evaluation of $n_s(0)$ in OD cuprates. For Tl2201 specifically, it confirms both the doping independence of $m^*$, even as the doping level $p^* = 0.19$ associated with the opening of the normal state pseudogap is approached, and the lack of in-plane anisotropy in $m^*$ or in $v_F$ throughout the doping series (otherwise Eq. (2) would not be valid). This lack of anisotropy is consistent with the expectation that the Fermi level in superconducting Tl2201 lies well above the van Hove singularity (vHs) [14,38].

Having established the doping dependence of $n_s(0)$, we next turn to examine the evolution of the coherent carrier density $n_{coh}$. The main result is summarized in Fig. 1B. The open triangles in Fig. 1B are the same $n_s(0)$ values transposed from Panel A. The black dashed line

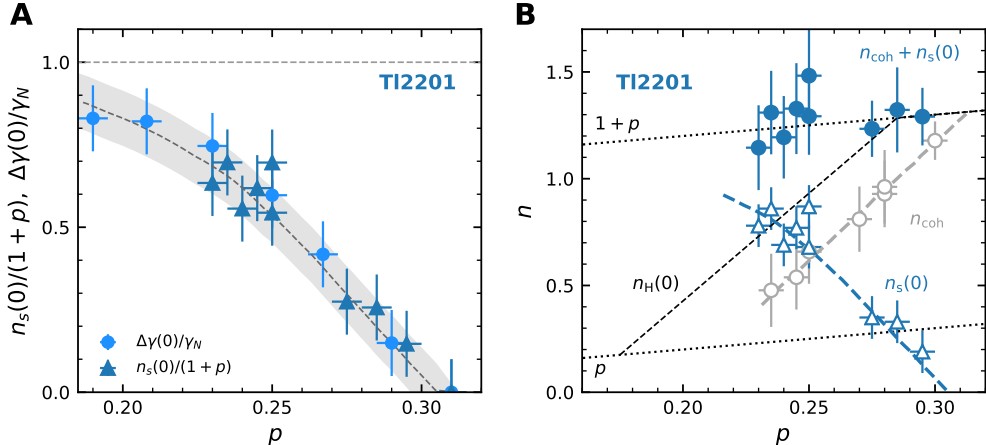

Figure 1: **Superfluid density and coherent normal-state carrier density in over-doped Tl2201. A)** Comparison of $n_s(0)/(1+p)$ and $\Delta\gamma(0)/\gamma_N$ in OD Tl2201, where $n_s(0)/(1+p)$ (triangles) is the fraction of the total Luttinger count $(1+p)$ that contributes to the as-measured superfluid density $n_s(0)$ [4–6,35] and $\Delta\gamma(0)/\gamma_N$ (circles) is the fractional change in the electronic specific heat coefficient in the superconducting state $\Delta\gamma(0)$ relative to its value $\gamma_N$ in the normal state [28,36] (see appendix A.7 for details). $n_s(0)$ is determined from $1/\lambda_{ab}^2(0)$ using Eq. (2) with $m^* = 5.2\, m_e$, consistent with $\gamma_N$ for OD Tl2201 across the doping series [28] (see appendix A.6 for details). The grey shaded region and dashed lines are guides to the eye. **B)** Open triangles: absolute $n_s(0)$ values (per Cu) transposed from Panel A. Black dashed line: schematic $n_H(0)$ line for Tl2201 (again normalised to a single Cu site) derived from high-field Hall effect measurements [14]. Open circles: corresponding $n_{coh}$ values (per Cu) for Tl2201, obtained by renormalising the as-measured $n_H(0)$ by the ratio (squared) of the coherent contribution to the total conductivity at 0 K, as determined by analysis of $\rho_{ab}(T)$ (see appendix A.1-3 for details). The faint dashed line is a fit through the data points. Blue filled circles: The sum $n_{coh} + n_s(0)$ for Tl2201. For each value of $n_s(0)$, the corresponding value of $n_{coh}$ was read off from the faint dashed line. The two thin dotted lines represent the relation $n = p$ and $n = 1 + p$. Estimates of the error bars for each set of data are described in detail in the appendices.

represents schematically the evolution of $n_H(0)$ across the strange metal regime in OD Tl2201 derived from high-field Hall effect measurements [14]. The anti-correlation between $n_s(0)$ and $n_H(0)$ in Fig. 1B is clear - as the Hall number increases toward $1 + p$, the superfluid density falls, at a similar rate, toward zero. While this finding is already striking and anomalous, it is $n_{coh}$ that is most relevant here, not $n_H(0)$. Recall that $n_H(0)$ is obtained directly from the measured Hall voltage via Eq. (1) and is not necessarily equivalent to $n_{coh}$, even in a system like Tl2201 with a single, cylindrical FS and an isotropic $v_F$. The modifying factor in this case is the presence of the second, incoherent sector deduced from the in-plane MR studies [16]. The precise way in which this second sector modifies the analysis of $R_H(0)$ depends on the nature of the sector and how this manifests itself in the transport properties. Here, we assume that both the coherent and incoherent sectors co-exist on the same (underlying) FS but are located at different regions in $k$-space, the former near the zone diagonals, the latter along the flat sections of FS near $(\pm\pi, 0)$ and $(0, \pm\pi)$. In this way, their contributions to the total conductivity are additive, as they would be in a normal two-fluid system. Secondly, we assume that the Hall conductivity $\sigma_{xy}$ of the incoherent sector is zero, as inferred in Ref. [14]. Based on these simplifying assumptions, $n_{coh}$ is found to be related to $n_H(0)$ via the following simple

expression:

$$n_{coh} = f_\sigma n_{\mathrm{H}}(0), \tag{3}$$

where $f_\sigma = (\sigma_{xx}^{coh}/\sigma_{xx}^{tot})^2$ is the square of the contribution of the coherent sector $\sigma_{xx}^{coh}$ to the total dc conductivity $\sigma_{xx}^{tot}$ (see appendix A.1 for full details). Since $f_\sigma \leq 1$, $n_{coh} \leq n_{\mathrm{H}}(0)$ for all dopings. Estimates of $f_\sigma$ for Tl2201 at various $p$ values, deduced from fitting the zero-field resistivity, are presented in appendix A.2. The resultant $n_{coh}$ values (per Cu) are plotted as open circles in Fig. 1B. Note that $n_{coh}$ appears to reach the $1 + p$ line at $p \sim 0.31$, i.e. where both superconductivity and the $T$-linear term in $\rho_{ab}(T)$ vanish [13].

In an ordinary BCS superconductor (in the clean limit), $n_s(0) = n_{coh}(\sim n_{\mathrm{H}}(0))$ as all (mobile) electrons condense into the superfluid. Introducing disorder drives the system towards the dirty limit and leads to a reduction in $n_s(0)$ through pair-breaking. A careful study of the variation of $n_s(0)$ as a function of doping in $SrTi_{1-x}Nb_xO_3$ demonstrated how $n_s(0)$ can decrease relative to $n_{\mathrm{H}}(0)$ due to a crossover from the clean to the dirty limit [39]. The evolution of $n_s(0)$ and $n_{coh}$ ($n_{\mathrm{H}}(0)$) in $SrTi_{1-x}Nb_xO_3$ and OD Tl2201 shows one crucial difference however. While $n_{coh}$ ($n_{\mathrm{H}}(0)$) in both cases increases with increasing doping beyond the optimal doping, the value of $n_{coh}$ in superconducting Tl2201 is always smaller than the total carrier density expected from hole counting. The different scenarios for strontium titanate and OD cuprates likely stem from the fact that the parent state is a band insulator in the former and a Mott insulator in the latter. Certainly, the observation that the sum $n_{coh} + n_s(0) \approx 1 + p$ (filled circles in Fig. 1B) is incompatible with any conventional BCS picture. This simple empirical relation is our central finding, one that does not rely on knowing the exact microscopic origin of the non-FL, strange metal component. Before discussing its implications, however, we first turn to consider whether a similar relation may also hold in another OD cuprate, namely $La_{2-x}Sr_xCuO_4$.

# 3 Superfluid and coherent carrier density in $La_{2-x}Sr_xCuO_4$

While the doping range of several cuprate families extends beyond $p^*$, only LSCO has been studied in sufficient detail to enable us to investigate how $n_{coh}$ and $n_s(0)$ evolve with doping across the entire strange metal regime, i.e. between $p^*$ and $p_{SC}$ – the doping level corresponding to the edge of the superconducting dome. Extracting both quantities in OD LSCO, however, is not as straightforward as it is in Tl2201, due to the distinct FS topology and strong in-plane anisotropies of the former. While in Tl2201, the FS remains hole-like far beyond $p_{SC}$ [14], the FS in LSCO undergoes a Lifshitz transition from hole-like to electron-like around $p = 0.195$ due to the Fermi level crossing the vHs at $(\pi, 0)$ [40]. As a result, the FS of OD LSCO contains not only significant anisotropy in $v_F$ (due to proximity of the Fermi level to the saddle point at $(\pi, 0)$), but also regions of electron- and hole-like curvature (see Fig. 2A for an illustration – note that the labels *inc* and *coh* do not necessarily correspond explicitly to the curvature). In addition, its impurity scattering rate $1/\tau_0$ (i.e., in the zero-temperature limit) is known to be highly anisotropic [41, 42] due to a predominance of small-angle scattering off impurities located outside of the $CuO_2$ planes [43]. The anisotropies in $v_F$ and $1/\tau_0$ thus conspire, rather than cancel, to produce a zero-temperature mean-free-path $\ell_0$ that can be up to two orders of magnitude larger at the zone edge than along the zone diagonals [42]. This extreme anisotropy, coupled with the change in curvature around the (in-plane) FS, means that the relation between $R_{\mathrm{H}}(0)$ and $n_{\mathrm{H}}(0)$ specified in Eq. (1) is no longer tenable. Hence, even within a picture based solely on coherent quasiparticles, there are specific details for LSCO that need to be taken into account before the possible influence of any incoherent states can be considered.

With this in mind, we set out to model self-consistently the as-measured transport and

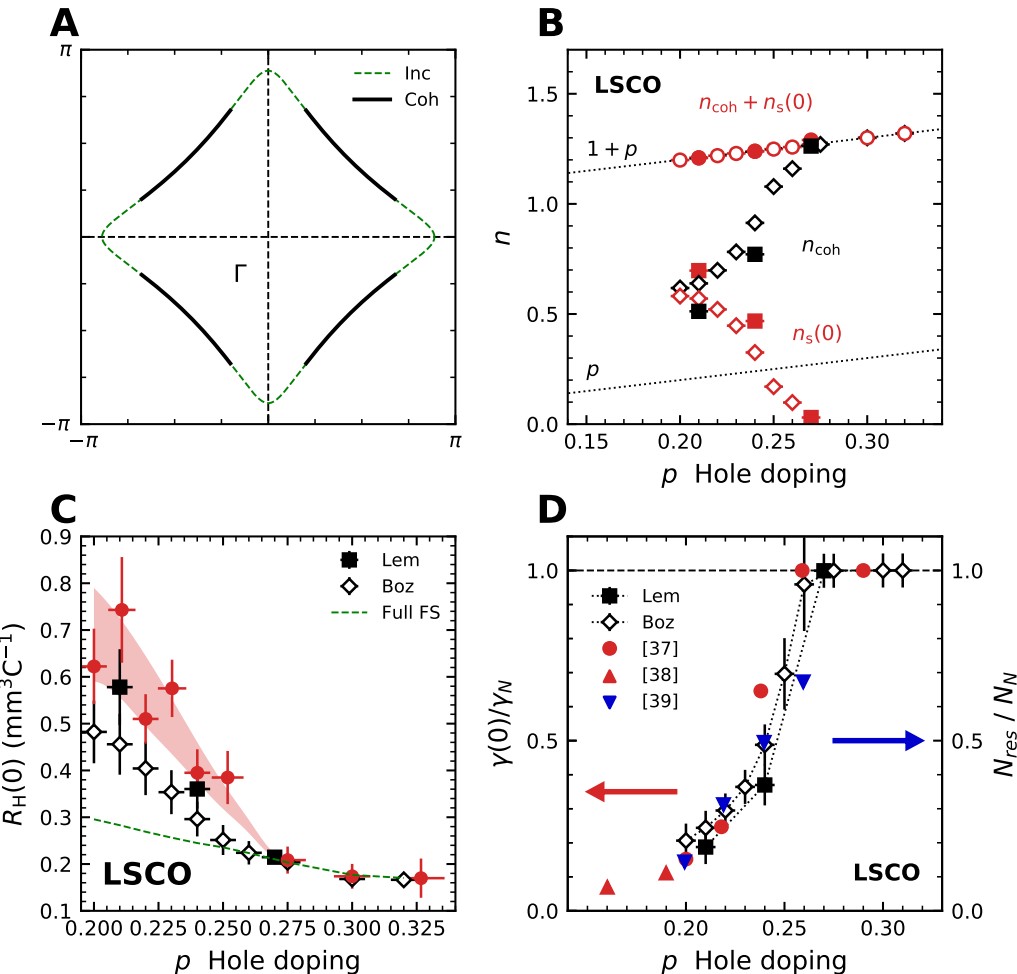

Figure 2: **Possible break-up of the Fermi surface in OD LSCO into coherent and incoherent sections and its possible consequences. A)** Example of a truncated FS (corresponding to $p = x = 0.23$) in which anti-nodal sections (green dashed lines) exhibit incoherent charge dynamics and the nodal sections (bold black lines) are coherent. Note that a similar delineation of the FS is seen in ARPES [44]. **B)** $n_s(0)$ at selected dopings extracted from mutual inductance measurements of $1/\lambda_{ab}^2(0)$ in LSCO thin films by Božović *et al.* (open red diamonds) [7] and Lemberger *et al.* (filled red squares) [45]. Black empty diamonds (filled squares) represent $n_{coh}$, the density of coherent carriers at low-$T$ deduced from each corresponding data set, assuming $n_{coh} = (1 + p) - n_s(0)$. **C)** Doping dependence of $R_H(0)$ in OD LSCO. (Red circles) Binned and averaged literature values for $R_H(0)$ (see Table 8). Red shaded area indicates the spread in the (binned and averaged) experimental values. Green dashed line represents the doping dependence of $R_H(0)$ calculated using Boltzmann transport theory and assuming the entire FS is coherent. Note the clear bifurcation between experimental and Boltzmann-derived values of $R_H(0)$ at $p_{SC} = 0.27$. Open diamonds (filled squares) represent the calculated $R_H(0)$ obtained by truncating the FS integral for $\sigma_{xy}$ as constrained by the $n_{coh}$ values shown in panel **B** and deduced from the Božović [7] (Lemberger [45]) $n_s(0)$ values. **D)** Residual density of states within the superconducting state of OD LSCO as determined by specific heat (circles [46] and triangles [47]) and Knight shift (inverted triangles) [48] studies. Again, open diamonds (filled squares) are the residual density of states calculated using the same truncated FS integrals employed in panel **C** and the $n_s(0)$ values deduced from Ref. [7] ( [45]), respectively. The error analysis for all panels is described in appendix B.

thermodynamic quantities of OD LSCO via the following strategy. We take as our starting point the two-dimensional (2D) tight-binding FS parameterization at various doping levels, as determined by angle-resolved photoemission spectroscopy (ARPES) [40]. We assume that any uncertainty introduced by ignoring the dispersion along $k_z$ is negligible, its effect being averaged out in any full 3D integral. We then use this 2D parameterization, along with the corresponding as-measured $v_F(\phi)$, to calculate an estimate for the normal state electronic specific heat coefficient $\gamma_N$ using the expression:

$$\gamma_N = \frac{V_{cell} k_B^2 N_A}{12\pi\hbar} \oint dS \frac{k_F(\phi)}{|v_F(\phi)|\cos\delta} \,. \tag{4}$$

Here, the units of $\gamma_N$ are J/mol·K$^2$, $\phi$ is the in-plane azimuthal angle, with $\phi = 0$ corresponding to the $k$-vector $(\pi, 0)$, and $\delta$ is the angle between $\mathbf{v_F}$ and the Fermi wave vector $\mathbf{k_F}$ (that ensures that the correct gradient is used for the calculation of $\gamma_N$). Taking $v_F(\phi)$ directly from ARPES, however, generates an estimate for $\gamma_N$ that is approximately one half of the experimentally-determined value [40]. The origin of this discrepancy is not yet known; it may reflect an additional low-energy renormalization in the quasiparticle dispersion – not detected by ARPES in LSCO, but seen, for example, in Bi2201 below 2−4meV [49] – or something more fundamental. In order to proceed, we simply scale the absolute value of $|v_F(\phi)|$ to match $\gamma_N$ for each representative doping level while maintaining its ARPES-derived $\phi$ dependence (since this appears to have been confirmed by different magneto-transport measurements [41,42]). The resultant scaling values are shown in Table 9 in appendix B.2. In this way, $\gamma_N(x)$ is taken to be the physical quantity against which all other quantities are determined. This choice is motivated by (i) the fact that measurements of $\gamma_N(x)$ in OD LSCO by different groups are in good agreement and show a systematic evolution with $x$ [46,50–52], and (ii) the expectation that both the coherent and incoherent sectors will contribute to the total density of states.

Thus, by combining specific heat and ARPES data, we are able to define, with reasonable confidence, $k_F(\phi)$ and $v_F(\phi)$ across the entire doping range of interest, i.e. $0.20 \le x, p \le 0.32$. The next step is then to compare the as-measured values of $1/\lambda_{ab}^2(0)$ [7,45] with those computed from the full integral formula:

$$\frac{1}{\lambda_{ab}^2(0)} = \frac{\mu_0 e^2}{4\pi^3\hbar} \oint dS \frac{v_x^2}{|v_F|\cos\delta} \,. \tag{5}$$

Here, $v_x$ is the $x$-component of $v_F$. The resulting comparisons are shown in Table 10 in appendix B.2. Clearly, as in OD Tl2201, only a fraction of the total states contribute to $n_s(0)$. At this stage, we make no assumption about which sector generates the superfluid in LSCO, but instead consider, on an equal footing, two scenarios $C(I)$ in which the superfluid emerges purely from coherent (incoherent) states, respectively. Taking our cue from the magneto-transport measurements [41, 42], however, we assume, as in Tl2201, that the coherent states are located near the 'nodal' regions of the Fermi surface (where the superconducting gap vanishes) while the incoherent states reside near $(\pi, 0)$. We then proceed by truncating the bounds of the integral in Eq. (5) at each $x$ until the calculated $1/\lambda_{ab}^2(0)$ matches the experimental value. Fig. 2A shows an example of such a truncation for $x = 0.23$. Intriguingly, the ARPES study of OD LSCO at $x = 0.23$ by Chang *et al.* [44] found that true FL quasiparticles (i.e. with an ARPES linewidth $\Gamma \sim \epsilon^2$) exist only around the nodal regions, while near the antinodes, $\Gamma \sim \epsilon$, signifying non-quasiparticle excitations. The resulting picture of anisotropic quasiparticle breakdown, including the extent of the FL-like and non-FL-like sectors on the Fermi surface, is strikingly similar to the one invoked here. The bounds themselves lead to a direct estimate of $n_s(0)$. Once $1/\lambda_{ab}^2(0)$ is matched, we then calculate $\gamma(0)$ and $R_{\rm H}(0)$ – *using precisely the same bounds* – and compare these with their corresponding experimental values

(where known). The full set of results for scenario $I$ are summarized in Fig. 2 (see appendix B for details, where scenario $C$ is also discussed).

Before comparing the outcomes from both scenarios, let us first elaborate on how $R_H(0)$ is calculated and then compared with experiment. As mentioned above, the variable FS curvature in OD LSCO gives contributions to $\sigma_{xy}$ of opposite sign [53]. This feature, combined with the violation of the isotropic-$\ell$ approximation, means that the simple Drude relation between $R_H(0)$ and $n_H(0)$ breaks down and the full Boltzmann expressions for both $\sigma_{xx}$ and $\sigma_{xy}$ [54] need to be considered. (The expressions themselves are described in appendix B.3.) Reassuringly, as shown in Fig. 2C, an excellent quantitative match to the experimental $R_H(0)$ values is found for dopings $p > p_{SC}$ using the full integrals. This agreement implies that the FS for non-superconducting OD LSCO is fully coherent, consistent with reports that the ground state of this region in the phase diagram is that of a correlated FL [52]. Below $p_{SC}$, however, the as-measured and FS-derived values for $R_H(0)$ appear to deviate from each other (the shaded region indicates the spread in $R_H(0)$ values from the literature - see appendix B.3). In this regime, the experimental $R_H(0)$ values can only be reconciled with the full integral formulae for $\sigma_{xx}$ and $\sigma_{xy}$ either by increasing $\delta$ (i.e. the effective FS curvature) or by increasing the anisotropy in $v_F(\phi)$ to values that are extreme and vastly different to those determined by ARPES [40] (again see appendix B.3 for details). One may also account for this change of slope in $R_H(0)(x)$, however, by invoking the presence of the two sectors below $p_{SC}$, to truncate the bounds of the FS integrals for both $\sigma_{xx}$ and $\sigma_{xy}$ (as done for $1/\lambda_{ab}^2(0)$ and $\gamma(0)$) and to set $\sigma_{xy} = 0$ in the incoherent sector (as done for OD Tl2201). Within scenario $I$, the region near $(\pi, 0)$ is the one which gives the positive (negative) contribution to $\sigma_{xy}$ ($R_H(0)$) [53]. Thus, by setting $\sigma_{xy} = 0$ in this region, the net effect is to make the calculated value of $R_H(0)$ more positive, thereby leading to a better match to the experimental value.

The black squares and open diamonds in Fig. 2C represent $R_H(0)$ values for OD LSCO obtained using the same FS parameterization and integral bounds that were used to match the $1/\lambda_{ab}^2(0)$ values at each respective $x$ (within scenario $I$). Good agreement between the calculated and experimental values is now found for all $x$. Note that in all cases, we have assumed $f_\sigma = 1$. This simplifying assumption is reasonable in LSCO given the large anisotropy in $\ell_0$ discussed above (see appendix B.4 for more details). The corresponding comparison for $\gamma(0)/\gamma_N$ is shown in Fig. 2D. Again, for scenario $I$, the overall trend is reproduced across the entire strange metal regime of OD LSCO. Hence, despite the very marked difference in maximum $T_c$, FS topology and disorder level in LSCO compared to Tl2201, the combined Hall density, superfluid density and specific heat data appear consistent with the same relation, i.e. $n_{coh} + n_s(0) = 1 + p$.

Finally, as discussed in appendix B.3, the alternative model (scenario $C$), with the condensate derived from near-nodal states, fails to give a consistent match between $n_s(0)$ and $\gamma(0)/\gamma_N$. Of course, pair breaking in a $d$-wave superconductor is expected to affect predominantly the nodal regions where the gap is smallest. Thus, the conjecture presented here is not necessarily a unique solution. Nevertheless, there are a number of other findings, discussed in appendix C, that appear to be in conflict with current predictions from dirty $d$-wave theory applied to OD LSCO. Moreover, the marked $x$-dependence of $R_H(0)$ in OD LSCO described above can only be modelled self-consistently within scenario $I$. The dirty $d$-wave model, by contrast, relies on the existence of a fully coherent FS for all $x$ across the strange metal regime.

# 4 Discussion

The anti-correlation between $n_s(0)$ and $n_{coh}$ outlined here implies that once superconductivity is suppressed, either by magnetic field or by temperature, the condensed carriers do not re-

emerge as coherent quasiparticles. Indeed, it is as though the superfluid condensate grows *at the expense of the coherent sector*. The total carrier density is of course fixed by the Luttinger count. The relation $n_{coh} + n_s(0) = 1 + p$ then becomes even more constraining, since it implies that the superconducting condensate in OD LSCO and Tl2201 derives only from those carriers that exhibit signatures of incoherent transport. It is important to remark that while all the key quantities: $n_s(0)$, $n_H(0)$, $\gamma(0)$ and $f_\sigma$, are defined in the zero temperature limit (either in zero magnetic field or in fields at which the superconductivity is suppressed), $f_\sigma$ (in Tl2201) has been deduced from finite-temperature resistivity curves (in LSCO, we have simply assumed that $f_\sigma \sim 1$). It is the (extrapolated) ratio of the zero-temperature (residual) resistivities of the two sectors that effectively renormalises $n_H(0)$ to obtain $n_{coh}$. Nevertheless, we do not expect the ratio of conductivities to vary significantly between 0 K and $T_c$ to affect these extrapolations.

The notion that superconducting coherence emerges out of an incoherent normal state in cuprates is not new [55]. Early ARPES studies showed that a coherent quasiparticle peak evolves below $T_c$ at the zone boundary (the so-called anti-nodal region) from an incoherent, normal-state background in UD and OP cuprates, and in addition, the bulk of the superconducting condensate originates from states near $(\pi, 0)$ [56, 57]. A marked enhancement in the microwave [58] or thermal conductivity below $T_c$ [59] also indicated a dramatic increase in the mean-free-path of uncondensed carriers, while in-plane optical studies of UD or OP $Bi_2Sr_2CaCu_2O_{8+\delta}$ (Bi2212) revealed that a significant fraction of the superfluid spectral weight comes from energies far above the superconducting gap scale ($4\Delta$) [60, 61]. All such features, however, were reported for UD or OP cuprates, and it has been largely assumed that such exotic features become weaker, or even vanish, on the OD side. Spectral weight transfer, for example, was found to be reversed in OD Bi2212, suggesting a possible recovery of conventional BCS condensation [62–64]. It was subsequently noted, however, that the vHs in Bi2212 (on one of the Fermi sheets) may markedly affect the proportionality between spectral weight transfer and the change in kinetic energy across $T_c$, potentially masking any intrinsic kinetic-energy saving [65].

How superconductivity evolves as the pseudogap opens below $p^*$ lies outside the scope of this work. Nevertheless, we can make some preliminary remarks here. It is well known that $n_s(0)$ in hole-doped cuprates peaks at $p^*$ and once the pseudogap opens, it drops precipitously [66]. At the same time, the signature of incoherent carriers in the normal state MR is lost and conventional [67] or modified [68] Kohler's scaling is recovered. These properties suggest that the incoherent carriers are predominantly gapped out below $p^*$ and as such are unable to contribute to the superfluid density. It has been largely assumed until now that the remaining states (i.e. on the Fermi arcs) also contribute to $n_s(0)$ in UD and OP cuprates. The presence of a residual electronic density of states (finite specific heat at $T = 0$) in even the very highest quality crystals [47, 69] and the emergence of quantum oscillations at magnetic field strengths below the solid-to-liquid vortex transition [70], may indicate otherwise.

Given the possible implications of our findings for the understanding of high-$T_c$ superconductivity, we conclude by highlighting some of its possible caveats. The scenario based on independent (parallel) conduction channels presented here assumes that the low-lying states associated with each sector are located in different regions in momentum space (in UD cuprates, this distinction is often referred to as the 'nodal-anti-nodal dichotomy'). While this is easy to conceptualize, it is not a unique interpretation. Moreover, it is not possible at this stage to determine whether a scenario in which the resistivities or the conductivities of the two sectors are added captures the experimental data better, even though the neatness of the obtained relation between $n_{coh}$ and $n_s(0)$ suggests that the parallel scenario is at least a viable starting point.

Secondly, the presence of incoherent states at the Fermi level seems, at first hand, incom-

patible with the observation of QO in OD Tl2201 [8]. It is important to note, however, that despite extensive efforts stretching over three decades, QOs have only been seen to date at very high dopings levels in Tl2201, i.e. beyond $p = 0.27$ (but still within the strange metal regime) [37]. At these doping levels, where the incoherent sector is envisaged to be small (and therefore possibly restricted in $k$-space), quasiparticles may experience the incoherent sector simply as an additional dephasing or modified Dingle term. In this regard, we note that the mean-free-path estimated from the Dingle analysis of QOs in heavily OD Tl2201 is roughly a factor of 2 shorter than that obtained from the in-plane resistivity (until now this discrepancy has been attributed to different contributions from small- and large-angle scattering). Alternatively, coherent quasiparticles may be able to traverse sufficiently narrow incoherent sections similar to how magnetic breakdown in a system with multiple pockets enables quasiparticles to tunnel across the breakdown gap. Both scenarios provide a mechanism by which QO can still be observed, provided the magnetic field is strong enough, though clearly QO studies on Tl2201 crystals doped across $p_{SC}$ would help to identify if there is indeed any additional dephasing present. It is worth noting too that QOs are a thermodynamic quantity and can arise in circumstances far departed from a conventional FL state. It has been shown theoretically, for example, that QOs can exist even in band-insulators [71] as well as in strongly interacting, quantum critical non-FLs [72]. Hence, the observation of QOs is no longer viewed as a 'smoking gun' for the existence of a fully coherent Fermi surface. Nevertheless, it remains a theoretical challenge to explain their existence inside the strange metallic state in OD cuprates.

Thirdly, one curious outcome of this analysis is the extended range of coexistence of the coherent and incoherent channels in OD Tl2201 relative to LSCO that echoes the extended range of superconductivity in the former. Within a dirty $d$-wave scenario for the cuprates, the reason for the extended range of superconductivity in Tl2201 on the OD side is obvious; lower disorder levels simply induce less pair-breaking. Indeed, it has been argued that both the variation of $n_s(0)$ with $T_c$ in LSCO (and Tl2201) and the $T$-dependence of $n_s(T)$ can be captured well by the dirty $d$-wave picture [73–75]. As shown in appendix C, however, the magnitude of the normal state scattering rate (obtained from the residual resistivity) in OD cuprates can be, when converted into units of temperature, more than one order of magnitude larger than the corresponding $T_c$ values. Such large values for the normal state scattering rate should, according to the predictions, extinguish all vestiges of superconductivity, irrespective of whether the scattering itself is in the Born or unitary limit. Moreover, variations in the residual resistivity appear to have little or no influence on $T_c$ itself, in marked contrast with expectations from the theory. In light of the analysis presented here, a re-examination of other claims of compatibility with the dirty $d$-wave scenario for OD cuprates may be timely.

Finally, a recent phenomenological model by Pelc *et al.* [76] has also invoked the coexistence of two sub-systems – one itinerant and FL-like, the other localized – across the OD superconducting regime of hole-doped cuprates; their sum recovering the full Luttinger count. As postulated here, the superfluid density in that picture is argued to derive from the localized, rather than itinerant carriers [76]. According to their model, the density of mobile carriers decreases continuously as $T$ falls below the effective (inhomogeneous) localization gap while their scattering rate maintains a strict $T^2$ dependence. Although similar in spirit to our own proposal, this picture is inconsistent with the observed drop in $R_H(T)$ at low $T$ and the quadrature scaling of the MR [16] which necessarily imposes a component in the zero-field resistivity with a $T$-linear (Planckian) scattering rate.

# 5 Conclusion

Taking as our motivation recent high-field magnetotransport measurements showing signatures of coherent and incoherent transport in the strange metal phase of overdoped cuprates, we have re-examined the London penetration depth, specific heat and Hall effect in overdoped $Tl_2Ba_2CuO_{6+\delta}$ and $La_{2-x}Sr_xCuO_4$. Our analysis reveals that the existing experimental data can be reconciled with a scenario in which these coherent and incoherent carriers are located on distinct regions of the underlying Fermi surface. Based on the assumption that the conductivities of these two sectors add in parallel, we have shown that in $Tl_2Ba_2CuO_{6+\delta}$, the growth of the superfluid density $n_s(0)$ with decreasing doping $p$ is quantitatively compensated with the decrease in the coherent carrier density $n_{coh}$ and that their sum is approximately equal to the full Luttinger count $1 + p$. Assuming a similar relation for $La_{2-x}Sr_xCuO_4$, we find good consistency in the evolution of the limiting low-$T$ Hall coefficient as well as the residual specific heat (inside the superconducting state). These correspondences, if confirmed, could indicate that, in contrast to the standard BCS theory, superconductivity in both $Tl_2Ba_2CuO_{6+\delta}$ and $La_{2-x}Sr_xCuO_4$ emerges from states that exhibit incoherent, rather than coherent transport in the normal state. Finally, the anti-correlation between $n_s(0)$ and $n_{coh}$ coupled with the non-FL transport properties exhibited right across the overdoped region, challenges previous notions that dirty $d$-wave BCS picture is an appropriate starting point for a description of OD cuprates. Evidently, OD cuprates need the strange metal phase in order to become superconducting.

# Acknowledgements

The authors would like to acknowledge stimulating discussions with M. Allan, I. Božović, M. S. Golden, B. Goutéraux, C. Pépin, K. S. Schalm, E. van Heumen and J. Zaanen.

**Funding information** We acknowledge the support of the HFML-RU/NWO, a member of the European Magnetic Field Laboratory (EMFL). This work is part of the research programme 'Strange Metals' (grant number 16METL01) of the former Foundation for Fundamental Research on Matter (FOM), which is financially supported by the Netherlands Organisation for Scientific Research (NWO). This work was also supported by the European Research Council (ERC) under the European Union's Horizon 2020 research and innovation programme (Grant Agreement No. 835279-Catch-22).

# A  Obtaining estimates for $n_{coh}$ and $n_s(0)$ in $Tl_2Ba_2CuO_{6+\delta}$

In a recent study, the low-$T$ Hall coefficient $R_H(0)$ of OD Tl2201 was measured for various doping levels in magnetic fields large enough to access the non-superconducting ground state [14]. At the lowest temperatures, the Hall resistivity $\rho_{xy}(H)$ was found to be linear in magnetic field strength $H$ and as a consequence, $R_H(0)$ was obtained from the asymptotic low-$T$ high-$H$ limit of $\rho_{xy}/H$, from which the low-$T$ Hall number $n_H(0) = V_{cell}/(R_H(0)e)$ was then determined. The evolution of $n_H(0)$ with doping is shown schematically as a faint dashed line in Fig. 1B.

In that work, it was assumed that $n_H(0)$ represented the number density of coherent charge carriers in OD Tl2201. A subsequent study of the in-plane magnetoresistance (MR), however, revealed evidence for incoherent carriers within the $CuO_2$ planes in addition to coherent quasi-particles [16]. Specifically, the magnitude of the MR was found to be at least one order of magnitude larger than expected, given the residual resistivity and corresponding (impurity) scattering rate. Its magnitude was also found to be insensitive to the orientation of the ap-

plied field, suggesting that the observed MR was non-orbital in nature. Moreover, the precise $H/T$ scaling seen in the MR implied that its origin was tied to that of the zero-field $T$-linear resistivity and associated with Planckian dissipation of the relevant carriers.

Collectively, these observations provide compelling evidence for the existence of two conducting sectors within the strange metal regime of OD Tl2201; one coherent, the other incoherent. Until now, however, it has not been possible to determine whether the conductivities of each sector add in parallel (as they would, for example, if they originated from different regions in $k$-space) or in series (e.g. if they originated from different scattering mechanisms). In this report, we consider the former and below, we show how within such a picture, the presence of the second (incoherent) sector modifies the determination of $n_{coh}$, the number density of coherent carriers, that contribute to the as-measured $\rho_{xy}(H)$ and $R_{\mathrm{H}}(0)$.

## A.1 Additive conductivity channels in Tl$_2$Ba$_2$CuO$_{6+\delta}$

Within a parallel conductivity scenario, we can write the total conductivity $\sigma_{xx}^{tot}$ as

$$\sigma_{xx}^{tot} = \sigma_{xx}^{coh} + \sigma_{xx}^{inc}, \tag{6}$$

where $\sigma_{xx}^{coh}$ and $\sigma_{xx}^{inc}$ are the coherent and incoherent components respectively. Similarly, the total Hall conductivity $\sigma_{xy}^{tot}$ can be decomposed into coherent ($\sigma_{xy}^{coh}$) and incoherent ($\sigma_{xy}^{inc}$) contributions, i.e.

$$\sigma_{xy}^{tot} = \sigma_{xy}^{coh} + \sigma_{xy}^{inc}. \tag{7}$$

After standard matrix inversion, the total Hall resistivity $\rho_{xy}^{tot}$ becomes a weighted sum of the coherent ($\rho_{xx}^{coh}$, $\rho_{xy}^{coh}$) and incoherent ($\rho_{xx}^{inc}$ and $\rho_{xy}^{inc}$) longitudinal and Hall resistivities respectively

$$\rho_{xy}^{tot} = \rho_{xy}^{coh} \frac{\left(\rho_{xx}^{tot}\right)^2 + \left(\rho_{xy}^{tot}\right)^2}{\left(\rho_{xx}^{coh}\right)^2 + \left(\rho_{xy}^{coh}\right)^2} + \rho_{xy}^{inc} \frac{\left(\rho_{xx}^{tot}\right)^2 + \left(\rho_{xy}^{tot}\right)^2}{\left(\rho_{xx}^{inc}\right)^2 + \left(\rho_{xy}^{inc}\right)^2}. \tag{8}$$

As mentioned above, the incoherent sector in OD Tl2201 exhibits an in-plane MR of a non-orbital origin [16]. Similarly, the evolution of $R_{\mathrm{H}}(0)$ in OD Tl2201 indicates that the Hall response from the incoherent sector is negligible [14], as indeed one might expect if orbital motion is impeded. With this in mind, we assume here that the contribution to the Hall effect from the incoherent sector is negligible, i.e.

$$\rho_{xy}^{inc} = 0, \tag{9}$$

and taking the approximation

$$\left(\rho_{xx}^{tot}\right)^2 \gg \left(\rho_{xy}^{tot}\right)^2 \tag{10a}$$

$$\left(\rho_{xx}^{coh}\right)^2 \gg \left(\rho_{xy}^{coh}\right)^2, \tag{10b}$$

as well as assuming a negligible change of the diagonal resistivity components in a magnetic field

$$\rho_{xx}^{tot} \approx \rho_{xx}^{tot}(H=0) = \rho_{xx,0}^{tot} \tag{11a}$$

$$\rho_{xx}^{coh} \approx \rho_{xx}^{coh}(H=0) = \rho_{xx,0}^{coh}. \tag{11b}$$

Eq. (8) can be simplified to

$$\rho_{xy}^{tot} \approx \rho_{xy}^{coh} \frac{\left(\sigma_{xx,0}^{coh}\right)^2}{\left(\sigma_{xx,0}^{tot}\right)^2}. \tag{12}$$

(The validity of approximations (10) and (11) will be discussed further in appendix A.5). Thus, even though the incoherent sector does not contribute to the Hall conductivity, the as-measured Hall resistivity must be rescaled by the weighting factor

$$f_\sigma = \frac{\left(\sigma_{xx,0}^{coh}\right)^2}{\left(\sigma_{xx,0}^{tot}\right)^2}, \tag{13}$$

in order to obtain $\rho_{xy}^{coh}$, the intrinsic Hall resistivity of the coherent sector. As will become clear, the weighting factor $f_\sigma$ plays a key role in determining $n_{coh}$.

The total ($\rho_{xy}^{tot}(0)$) and coherent ($\rho_{xy}^{coh}(0)$) zero-temperature Hall resistivities can be expressed as

$$\rho_{xy}^{tot}(0) = R_{\mathrm{H}}(0)\mu_0 H \tag{14a}$$

$$\rho_{xy}^{coh}(0) = R_{\mathrm{H}}^{coh}(0)\mu_0 H. \tag{14b}$$

Note that we have removed the suffix "*tot*" from $R_{\mathrm{H}}^{tot}(0)$ to be consistent with the labelling in the main text and to emphasize that $R_{\mathrm{H}}(0)$ is the as-measured Hall coefficient in the zero-temperature limit. In standard Drude notation, these Hall coefficients can be expressed as

$$R_{\mathrm{H}}(0) = \frac{V_{cell}}{n_{\mathrm{H}}(0)e} \tag{15a}$$

$$R_{\mathrm{H}}^{coh}(0) = \frac{V_{cell}}{n_{coh}e}. \tag{15b}$$

Thus, we infer that $n_{\mathrm{H}}(0)$ is directly obtained from measurements of $\rho_{xy}^{tot}(0)$. The conversion of the Hall resistivity into a carrier density may seem at first sight difficult to justify in cuprates – systems close to the Mott insulating state. However, Ando and co-workers measured the Hall response in lightly-doped LSCO and found that $n_{\mathrm{H}}(0) \sim x\,(p)$ for $0.01 < x(p) < 0.08$ [77]. Beyond $x = 0.08$, this correspondence breaks down, presumably due to the emergence of charge (stripe) order in the intermediate doping range $0.09 < x(p) < 0.16$. At high doping ($p > 0.27$), the relation $n_{\mathrm{H}}(0) \sim 1 + p$ has been confirmed in Tl2201. Hence, at both ends of the phase diagram, the relation between $n_{\mathrm{H}}(0)$ and the number of mobile holes appears to hold and it thus seems reasonable to assume that in the crossover regime $0.16 < p < 0.27$, the value of $n_{\mathrm{H}}(0)$ also provides a good estimate of the effective carrier density. Moreover, given that in Tl2201 the carrier density extracted from the Hall coefficient in the under- and far overdoped regimes corresponds to the known carrier densities of $p$ and $1 + p$, it is natural to assume that the Drude form of $R_{\mathrm{H}}(0)$ provides a measure of the carrier density across the entire range of doping. (Note that this simplified scenario does not apply to OD LSCO where the proximity to a vHs invalidates the assumption of an isotropic FS, as is discussed in Appendix B.)

This is not the full story, however. Because the incoherent sector affects $\sigma_{xx}^{tot}$ without contributing to $\sigma_{xy}^{tot}$, $\rho_{xy}^{tot}(0)$ does not represent the true coherent carrier density, even at high field and low temperature. Similarly, $n_{\mathrm{H}}(0)$ is an experimentally-derived quantity that does not reflect the actual coherent carrier density $n_{coh}$. The relation between the two is obtained by combining Eqs. (12)-(15)

$$n_{coh} = f_\sigma n_{\mathrm{H}}(0). \tag{16}$$

This is the same expression as that given in Eq. (3). Here, $n_{coh}$ represents the contribution to $\sigma_{xx}^{tot}$ and $\sigma_{xy}^{tot}$ from the coherent channel in Eq. (6). Note that similar reasoning does not hold

for the incoherent sector because, by assumption, it contributes only to $\sigma_{xx}^{inc}$, while there is no contribution to $\sigma_{xy}^{inc}$.

Since $f_\sigma < 1$ and $n_H(0)$ for $p < 0.27$ is either at or below the dashed line in Fig. 1B corresponding to the full Luttinger count, $n_{coh}$ must always be less than the total carrier density $n_{tot} = 1 + p$ whenever there is a finite incoherent sector present. To reflect this, we introduce a second factor $f_{coh}$ that represents the fraction of carriers that contribute to the coherent channel such that

$$n_{coh} = f_{coh} n_{tot} = f_{coh}(1 + p). \tag{17}$$

From charge conservation, we can write

$$n_{tot} = n_{coh} + n_{inc}, \tag{18}$$

where $n_{inc}$ represents the 'missing' charge that contributes to $\sigma_{xx}^{inc}$ but not to $\sigma_{xy}^{inc}$

$$n_{inc} = (1 - f_{coh})n_{tot}. \tag{19}$$

In this way, we establish how the incoherent sector manifests itself indirectly in both the Hall effect and resistivity data. In the following section, we proceed to fit the zero-field resistivity in order to obtain an estimate for $f_\sigma$ in OD Tl2201 for each doping level and from this, an estimate for $n_{coh}(p)$.

## A.2 Fitting the zero-field resistivity

Having introduced the weighting factors, we can now proceed to determine the separate contributions to the total (zero-field) conductivity in OD Tl2201. In total, five fitting parameters are required, two of them for the incoherent sector

$$\rho_{xx}^{inc}(T) = A + BT, \tag{20}$$

and three for the coherent sector

$$\rho_{xx}^{coh}(T) = C + DT + ET^2. \tag{21}$$

Note that both channels incorporate a finite residual resistivity in order to prevent $\rho(T = 0) = 0$. The linear $T$-dependence of $\rho_{xx}^{inc}(T)$ connects with the 'Planckian' quadrature MR reported in Ref. [16] and is assumed to be set by the Planckian dissipation limit

$$\frac{\hbar}{\tau} = \alpha k_B T, \tag{22}$$

where $\alpha$ is of order unity.

The expression for $\rho_{xx}^{coh}(T)$ contains both a Fermi-liquid quadratic term due to electron-electron scattering and an anomalous $T$-linear component. From analysis of the zero-field resistivity $\rho_{ab}(T)$ alone, it is difficult to ascertain whether the linear-in-$T$ component ($= DT$) in the coherent channel is required since it turns out to be the most sensitive of all 5 parameters. Indeed, in some cases, it is possible to obtain reasonably good fits to $\rho_{ab}(T)$ with $D$ set to 0. However, earlier analysis of $c$-axis angle-dependent magnetoresistance (ADMR) revealed the presence of two independent scattering channels in OD Tl2201; an isotropic $T^2$ scattering rate and an anisotropic $T$-linear scattering rate [78]. As this was derived from analysis based on Boltzmann transport theory, it is assumed to reflect the behavior of the coherent sector. A $T$-linear component of orbital origin was also found to govern the temperature and field dependence of the Hall resistivity $\rho_{xy}(H)$ using the same parameterization derived from the ADMR experiments (for a similar doping) [14]. Moreover, simultaneous fits of the in-plane MR

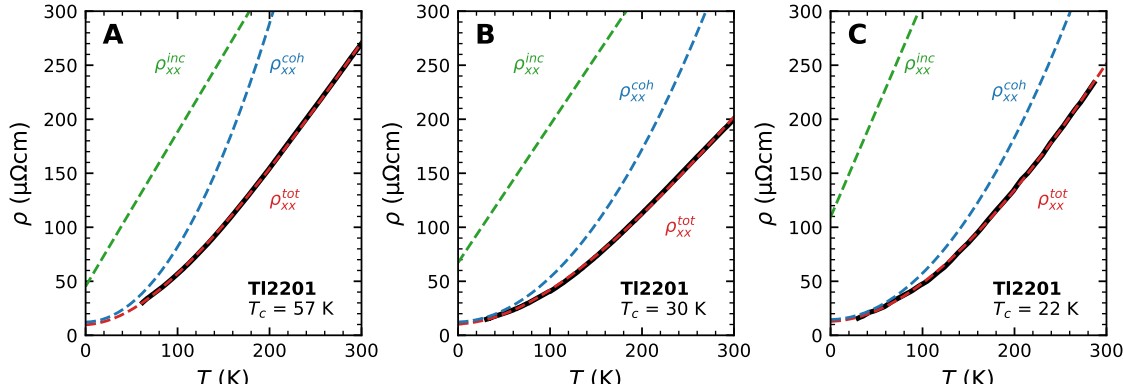

Figure 3: Parallel conductivity fits to the zero-field resistivities of OD Tl2201 with **A)**: $T_c = 57$ K [13], **B)**: $T_c = 30$ K [13] and **C)**: $T_c = 22$ K [16]. Black lines represent the measured data, the blue dashed lines are the $\rho_{xx}^{coh}(T)$ contributions described by Eq. (21) and the green dashed lines are the $\rho_{xx}^{inc}(T)$ contributions described by Eq. (20). The red dashed line represents the resultant fit to $\rho_{xx}^{tot}(T)$ as given by Eq. (23).

of OD Tl2201 measured at different temperatures [16] found the magnitude of $B$ to be smaller than the total $T$-linear term observed in the absence of a magnetic field, a further indication that a fraction of the total $T$-linear component in $\rho_{xx}^{tot}(T)$ originates from the coherent sector.

Finally, the experimentally-determined resistivity can be fitted to the following expression

$$\rho_{xx}^{tot} = \frac{\rho_{xx}^{coh}\rho_{xx}^{inc}}{\rho_{xx}^{coh} + \rho_{xx}^{inc}}, \tag{23}$$

which is nothing more than an inversion of Eq. (6). Examples are shown in Figure 3. Note that all fits were restricted to temperatures above which superconducting fluctuations were no longer evident. Typically, the zero-field resistivity was found to be insufficiently constrained to obtain fitted parameters that were insensitive to the initial fit conditions. However, some of the obtained values were unphysical (including negative residual resistivities in variables $A$ and/or $C$), and some were found to be inconsistent with other experimental evidence (e.g. fits lacking a finite $D$ are inconsistent with the ADMR results). Bounds were set on some of the parameters in order to constrain the overall fitting procedure. These bounds were as follows: $\rho_{upper} < A < 10$ mΩcm, $0.5$ μΩcm/K $< B$, $\rho_{lower} < C < \rho_{upper}$, $0.01$ μΩcm/K $< D$, $1$ nΩcm/K$^2 < E$, where $\rho_{lower}$ is the (extrapolated) zero-temperature limit of the as-measured resistivity and $\rho_{upper}$ is a multiple of $\rho_{lower}$ chosen to reflect the expected decrease in the contribution from the incoherent sector to the total resistivity with increased doping. This prevented unphysical fits, for example, ones in which a sample with a low incoherent carrier density had a small incoherent resistivity. The bounds on $A$ and $C$ ensure that the incoherent sector is always more resistive that the coherent sector. Finally, it is, of course, unphysical to assume that the electron-electron scattering term in the expression for $\rho_{xx}^{coh}(T)$ remains purely quadratic for all temperatures up to 300 K. However, since the Planckian term in Eq. 20 effectively 'saturates' the $T$-linear slope of $\rho_{xx}^{tot}(T)$ at high-$T$, its precise form is not expected to have any significant influence on the overall quality of the fits.

### A.3    Coherent carrier density in Tl$_2$Ba$_2$CuO$_{6+\delta}$

Since our focus here is on the *zero-temperature* limit of both the normal and superfluid carrier densities, the relevant weighting factors $f_\sigma$ are determined from the ratio of the residual

Table 1: Fitting parameters of the zero-field resistivities in the parallel conductivity channel scenario for Tl2201 [13, 16]. The parameters $A, B, C, D$ and $E$ are described in Eq. (20) and (21). The error bars indicate the standard deviation extracted from the fitting procedure. All parameters are also subject to an additional 20 % error resulting from the uncertainty in sample geometry that is not included here but is incorporated into the error analysis in Fig. 1.

| $T_c$ (K) | doping $p$ ($\pm 0.005$) | $A$ ($\mu\Omega$cm) | $B$ ($\mu\Omega$cm/K) | $C$ ($\mu\Omega$cm) | $D$ ($10^{-2}$ $\mu\Omega$cm/K) | $E$ ($10^{-3}$ $\mu\Omega$cm/K$^2$) | Ref. |
|---|---|---|---|---|---|---|---|
| 57 | 0.235 | $45 \pm 8$ | $1.43 \pm 0.05$ | $12.2 \pm 6.1$ | $1.0 \pm 0.3$ | $6.9 \pm 0.8$ | [13] |
| 48 | 0.245 | $36 \pm 2$ | $1.57 \pm 0.07$ | $10.3 \pm 0.5$ | $5.3 \pm 0.3$ | $5.1 \pm 0.2$ | [13] |
| 43 | 0.250 | $49 \pm 1$ | $1.61 \pm 0.13$ | $9.9 \pm 0.2$ | $1.0 \pm 0.2$ | $4.6 \pm 0.8$ | [13] |
| 30 | 0.270 | $67 \pm 7$ | $1.27 \pm 0.14$ | $12.2 \pm 1.0$ | $2.6 \pm 0.3$ | $3.9 \pm 0.4$ | [13] |
| 24 | 0.280 | $90 \pm 10$ | $1.20 \pm 0.05$ | $12.2 \pm 2.0$ | $2.4 \pm 0.2$ | $3.1 \pm 0.5$ | [16] |
| 22 | 0.280 | $109 \pm 5$ | $2.00 \pm 0.17$ | $15.5 \pm 6.7$ | $1.5 \pm 0.7$ | $4.1 \pm 1.7$ | [16] |
| 7 | 0.300 | $240 \pm 10$ | $26.1 \pm 2.2$ | $12.1 \pm 0.2$ | $1.3 \pm 0.4$ | $2.8 \pm 0.1$ | [13] |

resistivities using the following expression derived from Eqs. (6), (13), (20) and (21)

$$\frac{C}{A} = \frac{1 - \sqrt{f_\sigma}}{\sqrt{f_\sigma}} . \tag{24}$$

Note that as this is a ratio of resistivities, any uncertainty in the absolute resistivity values is removed. Having obtained $f_\sigma$ using Eq. (24), we proceed to determine an estimate of the coherent carrier density $n_{coh}$ for each doping level. The resultant values for $f_\sigma$ and $n_{coh}$ are listed in Table 2 and the doping dependence of $n_{coh}$ plotted as open circles in Fig. 4. The blue squares in Fig. 4 represent the Hall numbers $n_H(0)$ for OD Tl2201 as determined by Putzke *et al.* [14]. As can be seen, recognition of the presence of the incoherent sector has led to a reduction in our estimate of the coherent carrier density for all doping levels. Thus, where it would appear from measurements of $n_H(0)$ that the loss of coherent carriers begins at $p = 0.27$, after accounting for the presence of the second charge sector, $n_{coh}(p)$ now appears to extrapolate to $1 + p$ at $p \sim 0.31$, i.e., the doping at which superconductivity emerges.

## A.4 Planckian dissipation in Tl$_2$Ba$_2$CuO$_{6+\delta}$

Having obtained the contributions of the individual sectors to the total conductivity, we examine here the resistivity of the incoherent sector and in particular, the magnitude of the $T$-linear relaxation rate $1/\tau_{inc}$ associated with $\rho_{xx}^{inc}(T)$. As done previously [17, 79], we assume a Drude form for $\sigma_{xx}^{inc}$ and write

$$A + BT = \frac{m^*}{n_{inc} e^2 \tau_{inc}} . \tag{25}$$

Taking the derivative of Eq. (25) with respect to temperature and making use of Eq. (19), we find

$$\frac{d(\hbar/\tau_{inc})}{dT} = B(1 - f_{coh}) \frac{\hbar n_{tot} e^2}{m^* V_{cell}} , \tag{26}$$

where $V_{cell} = 173^3$ and $m^* = 5.2\, m_e$ (i.e., independent of doping), consistent with QO [37] and specific heat [28] measurements. (Note that the same approximation was used for determination of the superfluid density). The weighting factors $f_{coh}$ were determined using Eq. (17) and are listed in Table 3. In order to compare Eq. (26) with the Planckian expression (Eq. (22)),

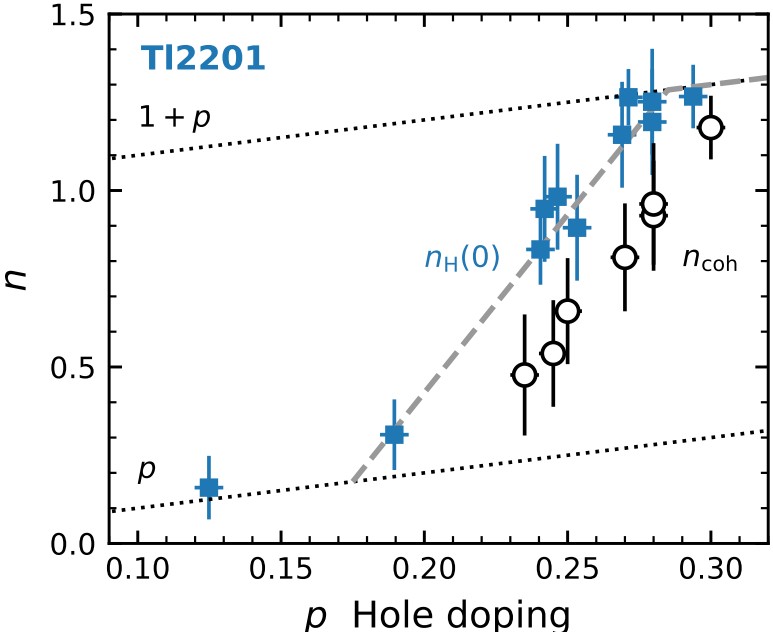

Figure 4: Measured Hall carrier density $n_H(0)$ in OD Tl2201 [14] together with the coherent carrier density $n_{coh}$ in the zero-temperature limit extracted using Eq. (16). In order to obtain $n_{coh}$, the Hall data [14] (blue squares) are first fitted to the faint dashed line ($n_H(0) = 0.175 + 10.1(p - 0.175)$). Then, for each $p$ value for which we have zero-field resistivity data, we obtain $f_\sigma$. Finally, multiplying the expression for $n_H(0)$ by these $f_\sigma$ values, we obtain the open symbols corresponding to $n_{coh}$. The error bars for $n_{coh}$ were calculated as composite deviations coming from standard deviations in $f_\sigma$ and in $n_H(0)$.

we set

$$\frac{d(\hbar/\tau_{inc})}{dT} = \alpha k_B . \tag{27}$$

Table 3 shows the resultant $\alpha$ values in Tl2201 for different doping levels. As we can see, for all the samples with $p < 0.295$, the parameter $\alpha$ is between 1 and 3, consistent with the value extracted in other cuprates [79] as well as many other correlated and quantum critical metals [17, 80]. Hence, even though the magnitude of the $T$-linear coefficient in OD Tl2201 implies a scattering rate much smaller than the Planckian limit [13], incorporation of the two charge sectors into the analysis of the zero-field resistivity reveals that the scattering rate associated with the incoherent sector is itself Planckian.

## A.5 Validity of approximations used in Section A.1

Here, we address the various approximations introduced in Section A.1 when fitting the zero-field resistivities to estimate the coherent and incoherent carrier densities. The first of these, introduced in Eq. (10), assumes that for each sample, the longitudinal resistivity ($\rho_{xx}^{tot}$ or $\rho_{xx}^{coh}$) far exceeds the corresponding Hall resistivity ($\rho_{xy}^{tot}$ or $\rho_{xy}^{coh}$). The second, expressed in Eq. (11), is that $\rho_{xx}^{tot} \approx \rho_{xx,0}^{tot}$ and $\rho_{xx}^{coh} \approx \rho_{xx,0}^{coh}$. Much of our analysis described above is based on fitting of the zero-field resistivity and in the zero-field limit, of course, all these approximations become exact. The key question, therefore, is whether the values of $n_H(0)$ extracted from the high-field Hall resistivity measurements of Ref. [14], are an accurate reflection of the $n_H(0)$ values one would obtain in the absence of superconductivity. In Figure 1 of Ref. [14], the

Table 2: Extracted coherent carrier densities in the zero-temperature limit for different doping levels in OD Tl2201. The measured Hall number $n_H(0)$ was determined for each doping level from the grey dashed line in Fig. 4 based on Ref. [14], while the weighting factor $f_\sigma$ was determined using Eq. (24) and the residual resistivity components listed in Table 1. Finally, Eq. (16) was used to obtain $n_{coh}$. The error bars for $f_\sigma$ were calculated as composite deviations coming from standard deviations in resistivity parameters $A$ and $C$ and the error bars for $n_{coh}$ were calculated as composite deviations coming from standard deviations in $f_\sigma$ and in $n_H(0)$.

| $T_c$ (K) | doping $p$ (±0.005) | $n_H(0)$ | $f_\sigma$ | $n_{coh}$ | Ref. |
|---|---|---|---|---|---|
| 57 | 0.235 | $0.77 \pm 0.10$ | $0.62 \pm 0.14$ | $0.48 \pm 0.17$ | [13] |
| 48 | 0.245 | $0.89 \pm 0.15$ | $0.60 \pm 0.02$ | $0.54 \pm 0.15$ | [13] |
| 43 | 0.250 | $0.95 \pm 0.15$ | $0.69 \pm 0.01$ | $0.66 \pm 0.15$ | [13] |
| 30 | 0.270 | $1.13 \pm 0.15$ | $0.72 \pm 0.03$ | $0.81 \pm 0.15$ | [13] |
| 24 | 0.280 | $1.21 \pm 0.15$ | $0.76 \pm 0.04$ | $0.93 \pm 0.16$ | [16] |
| 22 | 0.280 | $1.23 \pm 0.15$ | $0.78 \pm 0.08$ | $0.96 \pm 0.17$ | [16] |
| 7 | 0.300 | $1.30 \pm 0.09$ | $0.91 \pm 0.01$ | $1.18 \pm 0.09$ | [13] |

Table 3: Estimate of the prefactor $\alpha$ in the $T$-linear resistivity term associated with the incoherent sector. $B$ is copied from Table 1 and $f_{coh}$ is determined as described in the text. Parameter $\alpha$ is determined by comparing Eq. (26) and (27). The error bars for $f_{coh}$ were determined from the error bars in $n_{coh}$ and the error bars for $\alpha$ were calculated as composite deviations coming from standard deviations in $f_{coh}$ and the resistivity parameter $B$. The high error bars for the lower $T_c$ samples arise due to the low value of $(1 - f_{coh})$ used in Eq. (26) coupled with the uncertainty in $f_{coh}$.

| $T_c$ (K) | doping $p$ (±0.005) | $B$ ($\mu\Omega$cm/K) | $f_{coh}$ | $\alpha$ | Ref. |
|---|---|---|---|---|---|
| 57 | 0.235 | $1.43 \pm 0.05$ | $0.39 \pm 0.14$ | $2.78 \pm 0.64$ | [13] |
| 48 | 0.245 | $1.57 \pm 0.07$ | $0.43 \pm 0.12$ | $2.85 \pm 0.62$ | [13] |
| 43 | 0.250 | $1.61 \pm 0.13$ | $0.53 \pm 0.12$ | $2.45 \pm 0.65$ | [13] |
| 30 | 0.270 | $1.27 \pm 0.14$ | $0.64 \pm 0.12$ | $1.50 \pm 0.52$ | [13] |
| 24 | 0.280 | $1.20 \pm 0.05$ | $0.73 \pm 0.12$ | $1.08 \pm 0.48$ | [16] |
| 22 | 0.280 | $2.00 \pm 0.17$ | $0.75 \pm 0.14$ | $1.62 \pm 0.89$ | [16] |
| 7 | 0.300 | $26.1 \pm 2.2$ | $0.91 \pm 0.07$ | $8.1 \pm 6.1$ | [13] |

Table 4: Estimated superfluid density $n_s(0)$ (per Cu) in OD Tl2201 as a function of doping, derived from measurements of muon-spin relaxation [4, 5, 35] and microwave surface impedance [6]. The $T_c$ values are taken from the references. The corresponding $p$ values are obtained using the linear relation for $T_c(p)$ derived from quantum oscillation experiments [37]. Uncertainties for $p = 0.230, 0.235, 0.25(1), 0.25(2)$ are quoted in Ref. [6]; uncertainties for $p = 0.275$ are quoted in Ref. [35]. Errors for $p = 0.24$ and $0.295$ were obtained through the standard deviation of a linear fit to all four samples in Fig. 1 of Ref. [5]. Errors for $p = 0.245$ and $0.285$ were found through the uncertainty in the zero-temperature relaxation rate $\sigma_0$ fitting the data in Fig. 1(b) of Ref. [4] to the form $\sigma_0 \left[ 1 - (T/T_c)^\alpha \right]$ where $\alpha \leq 4$ [66].

| $T_c$ (K) | doping $p$ | $1/\lambda_{ab}^2(0)$ ($\mu$m$^{-2}$) | $n_s(0)$ | Ref. |
|---|---|---|---|---|
| 60 | 0.23 | $32.7 \pm 0.3$ | $0.78 \pm 0.01$ | [35] |
| 56 | 0.235 | $36.3 \pm 0.4$ | $0.86 \pm 0.01$ | [35] |
| 53 | 0.24 | $28.9 \pm 1.8$ | $0.69 \pm 0.04$ | [5] |
| 50 | 0.245 | $32.5 \pm 1.7$ | $0.77 \pm 0.04$ | [4] |
| 46 | 0.25(1) | $28.6 \pm 0.3$ | $0.68 \pm 0.01$ | [35] |
| 46 | 0.25(2) | $36.7 \pm 0.4$ | $0.87 \pm 0.01$ | [35] |
| 25 | 0.275 | $14.6 \pm 1.5$ | $0.35 \pm 0.04$ | [6] |
| 20 | 0.285 | $14.1 \pm 0.7$ | $0.33 \pm 0.02$ | [4] |
| 13 | 0.295 | $7.8 \pm 1.6$ | $0.19 \pm 0.04$ | [5] |

low-$T$ Hall coefficient $R_{\text{H}}(0)(= \rho_{xy}/H)$ in Tl2201 crystals with $T_c$ values of 30 K and 40 K is found to be independent of field from the maximum field strength (65 T) down to 20 T and 30 T respectively. Below these field scales, each sample enters the vortex regime. Given the evolution with field and temperature in each sample, together with the simulations presented in Ref. [14] based on Boltzmann transport analysis, we are confident that these $R_{\text{H}}(0)$ values are indeed representative of the low-$T$, low-$H$ Hall coefficients in OD Tl2201.

## A.6 Superfluid density of $\text{Tl}_2\text{Ba}_2\text{CuO}_{6+\delta}$

Table 4 lists the superfluid densities of OD Tl2201 samples measured via muon-spin relaxation ($\mu$SR) measurements on polycrystalline [4, 5] and single crystalline [35] samples and a microwave surface impedance measurement carried out on a Tl2201 single crystal with $T_c \approx 25$ K [6]. Excellent agreement is found between the various data sets. The doping levels quoted in Table 4 are obtained using the linear $T_c(p)$ dependence deduced from previous quantum oscillation studies [37], while the superfluid density values $n_s(0)$ are obtained using

$$n_s(0) = \frac{m^* V_{cell}}{\mu_0 e^2 \lambda_{ab}^2(0)}, \tag{28}$$

where $\lambda_{ab}(0)$ is the in-plane zero-temperature penetration depth and again, it is assumed that $m^* = 5.2\, m_e$ throughout. In order to obtain $1/\lambda_{ab}^2(0)$ for the Uemura data [4] where only values of the depolarization rate $\sigma$ were quoted, we used the relation from Niedermayer *et al.* [5]

$$\sigma[\mu s^{-1}] = 7.086 \cdot 10^4 \cdot 1/\lambda_{ab}^2(0)[\text{nm}]. \tag{29}$$

The resultant doping dependences of the superfluid, normal and total carrier densities in OD Tl2201 are presented in Fig. 1.

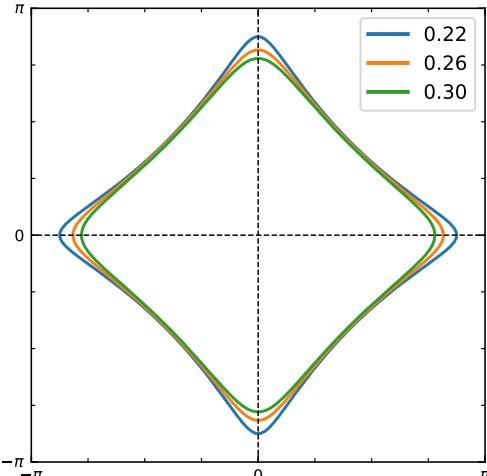

Figure 5: Tight binding parameterization for three dopings, $p = 0.22, 0.26, 0.30$ in OD LSCO based on ARPES spectra by Yoshida [40]. The FS contains both electron- and hole-like curvature which, coupled with an anisotropy in $\ell(\mathbf{k})$ at low-$T$, leads to contributions to $\sigma_{xy}$ of different sign.

## A.7    Electronic specific heat of $Tl_2Ba_2CuO_{6+\delta}$

The doping evolution of the electronic specific heat in OD Tl2201 has been determined on a single polycrystalline sample using differential calorimetry by Wade and co-workers [28] using a non-superconducting fully oxygenated reference sample of approximately the same weight. The use of a single sample minimised the influence of disorder, other than that originating from the interstitial oxygen, on the evolution of the residual specific heat (within the superconducting state). The $\gamma(T)$ data reported in Ref. [28] did not take into account entropy conservation above and below $T_c$. In Figure 6.9 of Ref. [36], the data are reanalysed using a small phonon correction in order to ensure entropy conservation. It is these values that are listed in Table 5. Here, $\gamma_N$ is the normal state electronic specific heat coefficient. In their work, Wade *et al.* found $\gamma_N = 6.7 \pm 1.1$ mJ/mol·K$^2$, independent of doping. Importantly, this doping independence of $\gamma_N$ is found to be robust to small changes in the phonon specific heat [36]. In our analysis, we fixed $\gamma_N = 7.4$ mJ/mol·K$^2$, consistent with values of the effective mass determined by quantum oscillation experiments [37]. Note that a similar analysis of this specific heat data, taking into account the requirement for entropy conservation, was also reported in Ref. [75].

## B    Obtaining estimates for $n_{coh}$ and $n_s(0)$ in $La_{2-x}Sr_xCuO_4$

In this section, we show how $n_{coh}$ and $n_s(0)$ are obtained for overdoped LSCO. The derivation of both requires more careful analysis than for Tl2201 due to the vicinity of the vHs in OD LSCO and the resultant strong anisotropy in the FS and in the low-$T$ mean-free-path $\ell_0(\mathbf{k})$. Nevertheless, LSCO remains to date the only cuprate beside Tl2201 for which sufficiently detailed Hall and superfluid density data are available to perform a similar analysis to that presented in Appendix A. Furthermore, these differences between LSCO and Tl2201 warrant a parallel study in order to verify whether or not the observed effects generalize across different cuprate families.

Table 5: Change in the electronic specific heat in OD Tl2201 upon entering the superconducting state as reported by Wade in Ref. [36]. The last column indicates the change in $\gamma$ normalised to its normal-state value $\gamma_N$. These ratio values are plotted in Figure 1A.

| $\delta$ | $T_c$ (K) | $p$ | $\Delta\gamma(0)$ (mJ/mol·K$^2$) | $\Delta\gamma(0)/\gamma_N$ |
|---|---|---|---|---|
| 0.001 | 85 | 0.190 | $6.7 \pm 0.5$ | $0.90 \pm 0.10$ |
| 0.017 | 77 | 0.208 | $6.1 \pm 0.5$ | $0.82 \pm 0.10$ |
| 0.032 | 61 | 0.230 | $5.5 \pm 0.5$ | $0.75 \pm 0.10$ |
| 0.052 | 46 | 0.25 | $4.4 \pm 0.5$ | $0.60 \pm 0.10$ |
| 0.063 | 32 | 0.267 | $3.1 \pm 0.5$ | $0.42 \pm 0.10$ |
| 0.077 | 17 | 0.290 | $1.1 \pm 0.5$ | $0.15 \pm 0.10$ |

Table 6: Tight-binding parameters for $La_{2-x}Sr_xCuO_4$ from Lee-Hone *et al.* [73]

| doping $p$ | $e_0$ | $t_0$ | $t_1$ | $t_2$ |
|---|---|---|---|---|
| 0.16 | 0.2025 | 0.2500 | $-0.0375$ | 0.0188 |
| 0.185 | 0.2080 | 0.2500 | $-0.0355$ | 0.0177 |
| 0.19 | 0.2096 | 0.2500 | $-0.0350$ | 0.0175 |
| 0.21 | 0.2144 | 0.2500 | $-0.0338$ | 0.0169 |
| 0.26 | 0.2295 | 0.2500 | $-0.0312$ | 0.0156 |

## B.1   Hall effect in $La_{2-x}Sr_xCuO_4$

In contrast to Tl2201, obtaining $n_H(0)$ in OD LSCO from measurements of the limiting low-$T$ Hall coefficient $R_H(0)$ is non-trivial. In this section, we address the pertinent issues and use the existing parameterization of the LSCO FS as well as knowledge of $\ell(\mathbf{k})$ to demonstrate why it is still necessary to invoke the presence of an incoherent channel in OD LSCO (more precisely, a region of the FS in which $\sigma_{xy} = 0$) to obtain a reliable estimate of $R_H(0)$ and from this, obtain estimates for $n_s(0)$ and $n_{coh}$ across the entire OD regime.

According to ARPES measurements, the quasi-2D FS in LSCO undergoes a Lifshitz transition around $x = p = 0.195$ when the Fermi level crosses the vHs near $(\pi, 0)$. Hence, beyond $x = 0.195$, the FS possesses both electron- and hole-like curvature, the former near the zone boundary and the latter near the zone diagonals (shown in Fig. 5 for three separate doping levels). A strong variation of the in-plane Fermi velocity $v_F$ then arises due to the presence of the closely-lying saddle points. In conventional metals, anisotropy in $v_F(\mathbf{k})$ is usually compensated for by anisotropy in $\tau(\mathbf{k})$ and as a result, $\ell_0$ becomes isotropic at zero temperature (the average distance between impurities within the conducting plane being independent of direction). In LSCO, however, it has been argued [43] that the scattering rate at a given momentum depends on the local density of states due to the fact that the (Sr) dopant impurities lie outside of the $CuO_2$ plane. This anisotropy in $1/\tau(\mathbf{k})$ in turn amplifies, rather than nullifies, any anisotropy in $v_F(\mathbf{k})$, leading to a marked violation of the isotropic-$\ell$ approximation. This violation then leads to a complicated expression for the low-field $R_H(0)$ which does not directly reflect the carrier density. The anisotropy in $\ell_0$, defined as the ratio $\beta = \ell(\pi, \pi)/\ell(\pi, 0)$, has been derived in both Hall effect [41] and ADMR [42] measurements and is found to be substantial, rising from $\beta \sim 10$ at $p = 0.33$ [41] to $\beta \sim 100$ at $p = 0.24$ (in Nd-LSCO) [42].

An elegant geometrical interpretation of the weak-field Hall conductivity $\sigma_{xy}$ in 2D metals was introduced by Ong in 1991 [53]. In metals with a FS that possesses both negative FS curvature (i.e. sections of opposing circulation of the $\ell$-vector as it is swept around the FS) and anisotropy in $\ell(\mathbf{k})$, $\sigma_{xy}$ is given by the integral of the 'Stokes' area $A = \int (d\ell \times \ell)/2$

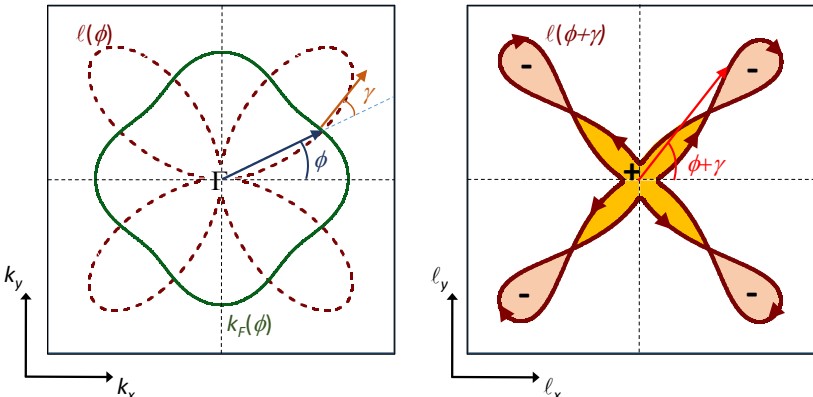

Figure 6: Left panel: Section of a 2D Fermi surface with pronounced negative curvature. The solid green line indicates the Fermi surface $k_F(\phi)$ and the brown dashed line indicates a strongly anisotropic mean free path $\ell(\phi)$. The blue arrow indicates the direction and length of $k_F$ while the orange arrow indicates the direction and length of $\ell$. Right panel: Polar plot of $\ell(\phi)$. The tangential arrows indicate the circulation of each loop and the $-/+$ signs indicate the corresponding sign of the loop. The red arrow again indicates the direction and length of $\ell$. The resultant $\sigma_{xy}$ is determined by the total area, i.e. the difference in the areas of the two counter-rotating loops.

over the full 2D FS. The corresponding '$\ell$-curve' contains areas with opposite circulation and therefore contributions to $\sigma_{xy}$ of opposite sign. A schematic example of this is given in Fig. 6 for a FS geometry similar to that realized in OD LSCO. The conductivity, $\sigma_{xx}$ is determined from the appropriate integral, and the corresponding Hall coefficient through $R_H = \sigma_{xy}/\sigma_{xx}^2$. Application of the Ong representation to heavily OD, non-superconducting LSCO ($x = 0.33$), using FS information derived from ARPES [40], was found to reproduce both $R_H(0)$ and its $T$-dependence up to 300 K [41]. Note that in this case, the full FS was included in the calculation of $R_H(0)$ while $\beta \approx 10$.

Table 7: Correction factor for the angle $\delta$ between $k_F$ and $v_F$; the uncertainty indicates the spread of values providing a Fermi surface which fits within the width of the ARPES resolution.

| doping $p$ | Correction Factor |
|:---:|:---:|
| 0.20 | $1.000 \pm 0.050$ |
| 0.21 | $1.005 \pm 0.050$ |
| 0.22 | $1.010 \pm 0.051$ |
| 0.23 | $1.025 \pm 0.051$ |
| 0.24 | $1.040 \pm 0.052$ |
| 0.25 | $1.055 \pm 0.053$ |
| 0.26 | $1.065 \pm 0.053$ |
| 0.275 | $1.075 \pm 0.054$ |
| 0.3 | $1.085 \pm 0.054$ |
| 0.32 | $1.115 \pm 0.056$ |

In order to explore whether the measured values of the Hall coefficient can be reproduced within the Ong representation for other dopings, we calculated $R_H(0)$ over the entire overdoped regime using the ARPES-derived FS parameterizations of Yoshida et al. [40]. (Note that all subsequent ARPES studies [44, 81–83] have found FS geometries that are consistent with the parameterizations reproduced in Fig. 5). It should be emphasized here that while the doping evolution of the FS appears slight, proximity to the vHs dictates that changes in the density of states and therefore in $\ell(\phi)$ around the FS have a strong doping dependence. Tight binding (TB) parameters are given in Table 6. The area enclosed by the TB-derived Fermi surfaces was found to correspond invariably to a FS of higher doping. Thus, a correction was made by increasing the Fermi wave vector $k_F$ until $\left(1 - 2 \times \frac{\text{Area FS}}{\text{Area BZ}}\right) = 1 + p$ where $p$ is the doping. To better fit the ARPES data of Yoshida et al. [40] for $p = 0.22$ and $p = 0.30$, the curvature of the FS was also altered by modifying slightly the angle $\delta$ between $k_F$ and $v_F$ while maintaining the correct Luttinger count. The multiplication factor was interpolated for intermediate dopings and the full set listed in Table 7. The strong FS curvature of LSCO is sensitive to slight changes in the tight-binding hopping parameters. Detailed studies [84, 85] of the 3D TB model for LSCO have included fourth-order in-plane hopping parameters and match well with LDA calculations at $k_z = 0$ (i.e the Fermi level). Indeed, the addition of this hopping parameter is found to alter the FS in the same direction as the correction factor for $p = 0.22$, though its doping dependence is not widely reported. Hence, here we restrict ourselves to the TB parameters given in [73]. Examples of uncorrected and corrected Fermi surfaces are shown for $p = 0.22$ and $p = 0.30$ in Figures 7B and 7C. The effect of these corrections was to increase slightly the modelled value of $R_H(0)$. For $p = 0.30$, the correction ensured an excellent match between the experimental and calculated Hall coefficient. It is important to note that variation in $\delta$ alone is not sufficient to fully renormalise $R_H(0)$ to the literature values for all $p < 0.27$. The green and blue dashed lines in Fig. 7A represent the evolution of the calculated $R_H(0)$ values based on the corrected and uncorrected FS parameterizations, respectively. The inverse scattering rate $\tau^{-1}(\mathbf{k})$ takes the Abrahams-Varma form [43] using the anisotropy of the TB-derived $v_F(\phi)$ multiplied by a constant chosen to obtain a residual resistivity $\rho_{xx}(0) = 20\ \mu\Omega$cm (a typical value in OD LSCO) in the fully coherent case. The anisotropy is reflected in $\ell(\mathbf{k})$ through $\ell(\mathbf{k}) = v_F(\mathbf{k})\tau(\mathbf{k})$.

Fig. 7 presents a summary of the experimentally-determined values for $R_H(0)$ in OD LSCO as reported in Ref. [41,77,86–92] together with others estimated from the maximum in $R_H(T)$. The reported values as well as the procedure for estimating $R_H(0)$ from $R_H(T)$ are presented in Table 8. The red shaded region reflects the spread in the (binned and averaged) literature values in OD LSCO. The green circles are the calculated $R_H(0)$ values and the green dashed line is an interpolation between these points. While for $p > 0.27$, the calculated $R_H(0)$ values assuming a full coherent FS agree extremely well with the as-measured values (in accordance with Ref. [41]), there is a clear bifurcation of the calculated and experimental values at $p = p_{SC} = 0.27$, i.e. precisely at the point where superconductivity emerges in heavily OD LSCO.

## B.2   Estimating the superfluid density in La$_{2-x}$Sr$_x$CuO$_4$

We begin by considering the residual electronic specific heat coefficient $\gamma(0)$ which is observed in the superconducting state but whose origin is not yet qualitatively understood [46, 94, 95]. Using Eq. (4) from the main text as well as the TB-derived FS parameters, we first calculate $\gamma_N$ (the normal state electronic specific heat coefficient) by integrating over the entire FS. As mentioned in the main text, there is a discrepancy of order 2 in the measured values relative to the calculated values. Table 9 shows the actual scaling parameter across the full doping range.

To proceed, we introduce the second (incoherent) charge sector as done in appendix A

Table 8: Experimental values for $R_H(0)$ for OD LSCO obtained from the literature. In samples where $T_c$ values were reported, the $p$ values were obtained using the standard parabolic $T_c(p)$ relation [93]. In others, the $p$ values are as given. Low-$T$ values $R_H(0)$, where reported, are listed. For those where only maximum $R_H$ values ($= R_H^{max}$) were plotted, $R_H(0)$ values were estimated by fitting the ratios $Z = R_H^{max}/R_H(0)$ of the other samples as a function of $p$, then using the (linear) fit to this ratio to estimate $R_H(0)^*$ from $R_H^{max}/Z_{fit}$. Using the same fit to 're-engineer' $R_H(0)$ for those samples for which $R_H(0)$ was already known shows that the fitting routine reproduced the as-measured $R_H(0)$ values to within 10 %. All extracted values for $R_H(0)$ and $R_H(0)^*$ were subsequently binned into $p$ steps of 0.01 (or higher) and plotted in Fig. 7.

| doping $p$ | $T_c$ (K) | $R_H(0)$ (mm³/C) | $R_H^{max}$ (mm³/C) | $Z = R_H^{max}/R_H(0)$ | $R_H^*(0) = R_H^{max}/Z_{fit}$ (mm³/C) | Ref. |
|---|---|---|---|---|---|---|
| 0.20 | | | 0.70 | | 0.72 | [86] |
| 0.20 | | 0.67 | 0.78 | 1.16 | 0.63 | [87] |
| 0.21 | 27 | 0.83 | 0.96 | 1.16 | 0.79 | [88] |
| 0.21 | | 0.60 | 0.78 | 1.29 | 0.63 | [89] |
| 0.21 | 26 | | 0.88 | | 0.59 | [90] |
| 0.22 | | 0.50 | 0.62 | 1.26 | 0.52 | [87] |
| 0.23 | 20 | 0.56 | 0.70 | 1.25 | 0.58 | [91] |
| 0.23 | | | 0.65 | | 0.54 | [77] |
| 0.23 | 21 | 0.60 | 0.70 | 1.17 | 0.58 | [88] |
| 0.235 | 18.5 | | 0.70 | | 0.39 | [90] |
| 0.24 | 16 | 0.34 | | | 0.34 | [88] |
| 0.24 | | 0.45 | 0.45 | 1.13 | 0.45 | [92] |
| 0.25 | | | 0.52 | | 0.44 | [77] |
| 0.25 | | 0.35 | 0.42 | 1.20 | 0.36 | [87] |
| 0.255 | 7.5 | | 0.46 | | 0.13 | [90] |
| 0.275 | 0 | 0.20 | 0.25 | 1.25 | 0.22 | [87] |
| 0.30 | 0 | 0.17 | 0.20 | 1.18 | 0.18 | [87] |
| 0.32 | 0 | | 0.14 | | 0.72 | [90] |
| 0.33 | 0 | 0.20 | 0.20 | 1.00 | 0.18 | [41] |

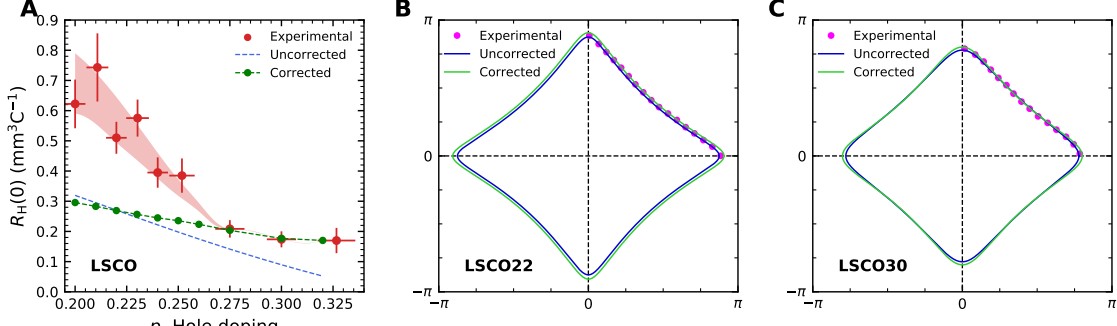

Figure 7: **A)**: Doping dependence of the low-$T$ $R_{\rm H}(0)$ in OD LSCO. (Red circles) Binned and averaged $R_{\rm H}(0)$ values obtained from the literature (see Table 8 and text for details of how these were obtained). The red shaded area provides an indication of the spread in the (binned and averaged) experimental values. The blue line is the evolution of $R_{\rm H}(0)$ estimated from Boltzmann transport theory using the TB parameterization of the (full) FS and assuming a scattering rate $1/\tau(\phi)$ with the same in-plane anisotropy as the ARPES-derived $v_F(\phi)$. The green circles are the values of $R_{\rm H}(0)$ calculated again assuming a fully coherent FS but now with a correction to satisfy the Luttinger count and adjust the FS curvature accordingly (but still remain consistent with the ARPES measurements to within their experimental momentum resolution). The green dashed line is an interpolation between these points. Note the bifurcation of the red symbols and the green dashed line at $p = p_{SC} = 0.27$. Error bars for the experimental $R_{\rm H}(0)$ values are obtained from the standard deviation of the binned literature values coupled with an estimated 10% uncertainty in the determination of the sample thicknesses. **B)**: Comparison between the uncorrected (blue) and corrected (green) Fermi surfaces for $p = 0.22$ compared with the locus (solid fuchsia circles) of the ARPES-derived FS from Yoshida *et al.* [40]. **C)**: Same comparison for $p = 0.30$. In both cases, the change in curvature matches well with the ARPES FS; for $p = 0.30$, the corrected and as-measured $R_{\rm H}(0)$ values are found to agree.

for Tl2201. We first adopt scenario $I$ and assume that the coherent states reside near the nodal points (i.e. where the $d$-wave superconducting order parameter vanishes) along $(\pi, \pi)$ while the incoherent states reside near the 'anti-nodes' near $(\pi, 0)$, as inferred from ARPES measurements within the strange metal regime [44]; the converse scenario (i.e. scenario $C$ with the incoherent states at the nodes) is considered at the end of section B.3. In order to simplify the subsequent calculations, the boundary between the two sectors is assumed to be sharp. In the following, we refer to the remaining coherent part of the FS as the truncated FS. Note that this truncation is conceptually different to what has been inferred previously for the pseudogap state, where, according to ARPES, the FS is truncated into disconnected Fermi arcs separated by regions where there is negligible spectral weight at the Fermi level. In our model, the states in those regions indicated by the green dashed lines in Figure 2A remain at the Fermi level (and thus contribute to $\gamma_N$, but nonetheless display incoherent non-FL transport behaviour).

Using Eq. (5), we first calculate $1/\lambda_{ab}^2(0)$ for the full FS (using the renormalised $v_F$ values) and truncate the FS integral until the calculation matches the as-measured value. The amount of truncation required then sets the coherent-incoherent boundary used in all subsequent calculations. Values for $1/\lambda_{ab}^2(0)$ and $n_s(0)$ are shown in Table 10. With the same coherent-incoherent boundary, $\gamma(0)$ is obtained from Eq. (4) from the appropriately truncated

Table 9: Renormalisation factors for the Fermi velocity of LSCO where $v_F^0$ is taken from ARPES-derived tight-binding Fermi surfaces [40] and $v_F$ is the Fermi velocity required to match the as-measured normal-state electronic specific heat $\gamma_N$ [46, 50–52].

| doping $p$ | $v_F/v_F^0$ | Reference |
|---|---|---|
| 0.20 | 1.96 | [7] |
| 0.21 | 2.27 | [45] |
| 0.21 | 2.27 | [7] |
| 0.22 | 2.46 | [7] |
| 0.23 | 2.56 | [7] |
| 0.24 | 2.60 | [45] |
| 0.24 | 2.60 | [7] |
| 0.26 | 2.53 | [7] |
| 0.27 | 2.46 | [45] |
| 0.30 | 2.15 | — |
| 0.32 | 1.71 | — |

Table 10: Measured and calculated $1/\lambda_{ab}^2(0)$ values for a full and truncated FS in OD LSCO. $n_s(0)$ is the estimated superfluid density (per Cu site) obtained as described in the text. For $p = 0.27$, the model was unable to reach the experimentally determined penetration depth. Thus, $n_s(0)$ is calculated using the London equation with $m^* = 8.2$ where $m^*$ is estimated from the as-measured electronic specific heat [7, 45]. Obviously, no penetration depth results have been reported on the non-SC samples but they are included here for completeness.

| doping $p$ | $1/\lambda_{ab}^2{}^{meas}(\mu m^2)$ | $1/\lambda_{ab}^2{}^{full}(\mu m^2)$ | $1/\lambda_{ab}^2{}^{trunc}(\mu m^2)$ | $n_s(0)$ | Reference |
|---|---|---|---|---|---|
| 0.20 | 16.70 | 69.67 | 16.67 | 0.581 | [7] |
| 0.21 | 20.30 | 59.63 | 20.25 | 0.697 | [45] |
| 0.21 | 14.90 | 59.63 | 14.85 | 0.570 | [7] |
| 0.22 | 12.60 | 54.19 | 12.55 | 0.521 | [7] |
| 0.23 | 10.20 | 52.03 | 10.18 | 0.447 | [7] |
| 0.24 | 11.10 | 50.97 | 11.06 | 0.468 | [45] |
| 0.24 | 7.00 | 50.97 | 6.99 | 0.325 | [7] |
| 0.25 | 3.60 | 51.43 | 3.61 | 0.170 | [7] |
| 0.26 | 0.70 | 51.83 | 0.68 | 0.098 | [7] |
| 0.27 | 0.15 | 51.89 | 0.00 | 0.030 | [45] |
| 0.3 | 0.00 | 59.39 | 0.00 | 0.000 | — |
| 0.32 | 0.00 | 73.39 | 0.00 | 0.000 | — |

integral. The resultant ratios $\gamma(0)/\gamma_N$ are plotted in Fig. 2D and compared with the experimental data summarized in Ref. [46, 47] as well as with the Knight shift results of Ref. [48]. A scenario in which the FS is divided into two distinct charge sectors is thus found to account well for the observed doping dependence of the uncondensed carriers.

## B.3 Coherent carrier density in La$_{2-x}$Sr$_x$CuO$_4$

Having established the degree of truncation of the FS to match $1/\lambda_{ab}^2(0)$ to experiment, we now return to the discrepancy between the as-measured Hall coefficient and the modelled

values for the fully coherent FS revealed in Fig. 7A. We again proceed by assuming that the incoherent sector has a longitudinal conductivity that is additive but a Hall conductivity that is zero. We then apply the Ong construction for $\sigma_{xy}$ but with the integral truncated by the boundaries defined in section B.2. The precise expression for $\sigma_{xy}$ (with the full integration limits) is [54]:

$$\sigma_{xy} = -\frac{e^2 \mu_0 H}{2\pi^2 \hbar^2 d} \int_0^{2\pi} \ell_x(\phi) \frac{\partial}{\partial \phi} \ell_y(\phi) d\phi \, . \tag{30}$$

The variation in $\ell(\phi)$ and in $k_F(\phi)$ used in the above expression is determined as described above. All symbols have their usual meanings, $\mu_0 H = 1$ T, and $d = 6.6$ Å is the interplanar distance. The fraction of coherent FS, $f_{coh}$, is simply given by the ratio $1 - n_s(0)/(1 + p)$. The calculated $R_H(0)$ values are listed in Table 11 and plotted in Figures 2B and 2C, respectively. As can be seen in Figure 2C, the truncated values of $R_H(0)$ are within 0.2 mm$^3$/C of the experimental value and the distinct bifurcation from the full FS calculation at $p_{SC} = 0.27$ is well captured by the simulation.

In order to test the validity of our assumption that incoherent transport derives from carriers located at the anti-nodal regions of the FS, the above calculations were repeated with inverted integral boundaries (scenario $C$), i.e. the coherent and incoherent sectors positioned at the anti-nodes and nodes respectively. For all dopings within the superconducting dome, $\gamma(0)/\gamma_N$ showed no increase with doping and the resulting $R_H(0)$ became negative (the region of the FS with positive curvature now having been truncated out) with a value that is one order of magnitude larger than the experimental values. In addition, the calculated incoherent carrier densities conflicted with values inferred from superfluid density measurements, as shown in Table 12. Thus, 'scenario $C$' was deemed to be inappropriate for OD LSCO.

On empirical grounds, one could further imagine that incoherence affects $\sigma_{xy}$ isotropically. In order to explain an increase in $R_H(0) = \sigma_{xy}/\sigma_{xx}^2$ by a factor 2-3, as indicated in Fig. 2C, a sizable drop in $\sigma_{xx}$ would be required. However, such a decrease is not observed in the model where $\sigma_{xx}$ remains comparatively unaltered. Thus, an anisotropic change in the Hall conductivity is required. Combined with the above, we conclude that preferential suppression of $\sigma_{xy}$ near the anti-nodal regions best captures the observed enhancement in $R_H(0)$ with reduced doping, in line with scenario $I$.

## B.4 The weighting factor $f_\sigma$ in La$_{2-x}$Sr$_x$CuO$_4$

As described in section A.1, in a model based on coherent and incoherent conductivity channels summing in parallel, $R_H(0)$ has to be renormalised by the weighting factor $f_\sigma$ in order to obtain an estimate for the coherent carrier density $n_{coh}$. The weighting factor $f_\sigma$ is equal to the square of the ratio between the zero-field conductivity of the coherent channel and the total zero-field conductivity (Eq. (13)) and can be determined by fitting the total zero-field resistivity, as described in section A.2 for Tl2201. For full consistency, therefore, the same weighting factor $f_\sigma$ should also be included when extracting the coherent carrier density from the measured Hall coefficient in LSCO. In contrast to Tl2201, however, it has proven impossible to perform a reliable 5-parameter fitting procedure on the zero-field resistivity in LSCO to extract $f_\sigma$. This difficulty likely stems from the fact that $\ell_0$ can be 1 to 2 orders of magnitude smaller at the zone boundary than along the zone diagonals, even if the entire FS of OD LSCO were coherent. Taking into account any additional scattering, e.g. on critical fluctuations, that could drive states at the zone edges incoherent, the contribution to $\sigma_{xx}$ from those states would become even smaller. In light of this, it would appear that the total conductivity in the zero-temperature limit comes almost entirely from the coherent sector, i.e. that $f_\sigma \approx 1$. Thus, in our analysis of $R_H(0)$, we have assumed that $f_\sigma = 1$.

Table 11: Measured Hall coefficient $R_H^{meas}(0)$ in the zero-temperature limit (based on Table 8) for OD LSCO together with $R_H^{full}(0)$ – the calculated Hall coefficient assuming a fully coherent FS, and $R_H^{trunc}(0)$ – the calculated Hall coefficient for a truncated FS within scenario $I$ determined using $n_{coh} = (1 + p)$ - $n_s(0)$ and the $n_s(0)$ values listed in Table 10.

| doping $p$ | $R_H^{full}(0)$ (mm$^3$/C) | $R_H^{meas}(0)$ (mm$^3$/C) | $R_H^{trunc}(0)$ |
|---|---|---|---|
| 0.20 | 0.296 | 0.69 ± 0.09 | 0.482 ± 0.067 |
| 0.21 | 0.283 | 0.65 ± 0.07 | 0.578 ± 0.081 |
| 0.21 | 0.283 | 0.65 ± 0.07 | 0.456 ± 0.065 |
| 0.22 | 0.269 | 0.58 ± 0.07 | 0.404 ± 0.057 |
| 0.23 | 0.257 | 0.49 ± 0.06 | 0.354 ± 0.047 |
| 0.24 | 0.249 | 0.34 ± 0.04 | 0.360 ± 0.042 |
| 0.24 | 0.249 | 0.40 ± 0.06 | 0.296 ± 0.037 |
| 0.25 | 0.237 | 0.35 ± 0.02 | 0.251 ± 0.032 |
| 0.26 | 0.228 | 0.28 ± 0.01 | 0.224 ± 0.025 |
| 0.27 | 0.217 | — | 0.215 ± 0.007 |
| 0.275 | 0.205 | 0.22 | 0.204 ± 0.006 |
| 0.3 | 0.172 | 0.21 | 0.168 ± 0.006 |
| 0.32 | 0.170 | 0.19 | 0.166 ± 0.006 |

Table 12: Calculated $n_s(0)$, $n_{coh}$ and $R_H(0)$ for scenario $C$ in which the incoherent carriers occupy the nodal regions of the Fermi surface. The large negative Hall coefficients are clearly inconsistent with the literature. Moreover, the lack of a strong doping dependence in $\gamma(0)/\gamma_N$ fails to account for the measured residual specific heat in OD LSCO.

| doping $p$ | $n_{coh}$ | $n_s(0)$ | $\gamma(0)/\gamma_N$ | $R_H(0)$ (mm$^3$/C) |
|---|---|---|---|---|
| 0.20 | 0.54 | 0.23 | 0.48 | −4.41 |
| 0.21 | 0.54 | 0.24 | 0.48 | −4.43 |
| 0.22 | 0.55 | 0.23 | 0.48 | −4.23 |
| 0.23 | 0.57 | 0.19 | 0.49 | −3.97 |
| 0.24 | 0.59 | 0.14 | 0.49 | −3.57 |
| 0.26 | 0.63 | 0.00 | 0.50 | −2.82 |

Having established that $f_\sigma \approx 1$, a natural question arises: if the total conductivity is dominated by the coherent sector, why does the zero-field resistivity of OD LSCO vary almost linearly with temperature over such a wide doping range [12]? In order to address this question, we first recall that the total conductivity in LSCO is dominated by the coherent sector only in the zero-temperature limit, where the impurity scattering is not screened by other scattering mechanisms. With increasing temperature, when other scattering mechanisms come into play, it is expected that the coherent sector, whose resistivity grows faster as $T + T^2$ (see section A.2), is no longer dominating the total conductivity entirely. Such behavior can also be seen in Tl2201 (see Fig. 3) where, at low $T$, the coherent sector gives the dominant contribution, while at higher temperatures, the mixing between the two channels produces the total resistivity that shows an almost $T$-linear dependence on approaching 300 K. In case of LSCO, such a mixing is expected to occur at much lower temperatures, meaning that the coherent sector grows much faster than in case of Tl2201, which, based on zero-field resistivity alone, is impossible to determine.

## B.5 Additional considerations for La$_{2-x}$Sr$_x$CuO$_4$

While the data plotted in Fig. 2 appears to show that the relation $n_{coh} + n_s(0) = 1 + p$ holds equally well in both OD Tl2201 and LSCO, we conclude this section by considering here other factors that have been ignored until now and that may influence the final robustness of the posited relation.

The first point of consideration is that all calculations of $\sigma_{xx}$, $\sigma_{xy}$ and $n_s(0)$ were performed for a strictly 2D FS. While this is likely to be a good approximation for OD T2201, where the resistivity anisotropy is more than 1000 [96], it is not immediately clear whether the approximation holds as well in OD LSCO, where the resistivity anisotropy becomes less than 50 [52]. It is not known at present how these calculations will be modified by inclusion of a finite $c$-axis FS warping, though we expect any modifications due to the warping to be effectively averaged out in a full 3D integration.

The second point of consideration is the difference between single crystals and thin films. While the majority of the analysis has been performed on transport and thermodynamic data obtained on bulk single crystals, all $n_s(0)$ values were obtained from penetration depth measurements carried out on thin films. Strain from the substrate is known to modify the properties of LSCO, but it is not at all clear how to factor this into the calculations. In the work by Lemberger *et al.* [45], for example, two of their films – with nominal $x$ values of 0.27 and 0.30 – are found to have $T_c$ values of 21 K and 9 K. We did not use these samples in our analysis as their $T_c$ values are far from the expected (Presland) parabola [93] and their corresponding $1/\lambda_{ab}^2(0)$ values markedly different from the values quoted by Božović *et al* [7]. At the same time, it is noted that superconductivity in the Božović films vanishes at a Sr concentration of 0.26, while in single crystals, $p_{SC} = 0.27$. Such a small shift in the range of superconductivity, however, will only modify the analysis slightly and is not expected to affect any of the main conclusions.

## C  Dirty $d$-wave scenario in overdoped cuprates revisited

Following the reports on the anomalous superfluid density [7] and optical conductivity [97] in OD LSCO, a dedicated theoretical study was carried out seeking to explain such behavior within a dirty $d$-wave scenario based on weak-coupling BCS theory [73–75]. Specifically, the effects of impurity scattering were taken into account through a self-consistent T-matrix approximation (SCTMA) and found to reproduce both the magnitude of the superfluid density and its (predominantly linear) dependence on temperature, provided that the vast majority of impurity scatterers were in the Born limit. In this section, we confirm the parameterization used in these calculations is consistent with that obtained from transport studies but show that the same theory fails to account for the apparent insensitivity of $T_c$ to the absolute value of the residual resistivity $\rho_0$ in OD cuprates.

The key parameter in the SCTMA analysis is $\Gamma_N(0)$, the zero-temperature normal state scattering rate. Disorder leads to a closing of the energy gap at a reduced $T_c$, the reduction being set by the celebrated Abrikosov-Gorkov formula [98]. Importantly, $T_c$ is found to depend only on $\Gamma_N/T_{c0}$, while the form of $\rho_s(T)$ is influenced heavily by the impurity phase shift. Lee-Hone *et al.* [73] showed that within the Born limit, $\rho_s(T)$ remains $T$-linear at the lowest temperature even for $\Gamma_N(0) = 0.5\ T_{c0}$, despite the fact that both $\rho_s(0)$ and $T_c$ have been reduced by approximately 40 %.

The scattering rate relevant for determining the drop in superfluid density is the elastic scattering rate set by the residual resistivity $\rho_0$. As done by Lee-Hone *et al.* [73], the resistivity

is obtained from the Drude formula:

$$\rho_0 = \frac{1}{\sigma_0} = \frac{m^*}{ne^2}\Gamma_{tr}(0), \tag{31}$$

where $m^*$ is the effective mass, $n$ is the carrier density and $\Gamma_{tr}(0)$ is the transport relaxation rate. The carrier density is estimated simply from the Sr content $x$ with $n = (1+x)/V_{cell}$. For $x = 0.24$, we obtain $m^* \approx 8\, m_e$ from the low-$T$ electronic specific heat [99]. For $\rho(0) \approx 16\,\mu\Omega$cm, one then obtains $\Gamma_{tr}(0) \approx \Gamma_N(0) \approx 55$ K, in agreement with Lee-Hone *et al.* [100]. Taking into account the FS geometry, the anisotropy in the Fermi velocity and in the lifetime $\tau_{tr}(\phi)$ deduced from Hall effect measurements [41], produces only a 10 % variation in this estimate of $\Gamma_N(0)$. Note, however, that this estimate of $\Gamma_{tr}(0)$ is a lower limit since the true value of $m^*$ in this formula is more likely to be closer to the unrenormalized or band mass value.

The key feature of Ref. [73] is the strong sensitivity of $T_c$ to the value of $\Gamma_N(0)$. Lee-Hone *et al.* found, for example, that for a $\Gamma_N(0)$ of this magnitude (more precisely, a residual resistivity of 16 $\mu\Omega$cm), the transition temperature of an OD LSCO film with $x = 0.25$ is suppressed from its clean-limit value of 65 K to 10 K. This strong sensitivity of $T_c$ to the value of $\Gamma_N(0)$ implies that samples with different residual resistivities should have markedly different $T_c$ (and $\rho_s$) values. Zn is known to have a strong detrimental effect on $T_c$ [101], other dopants less so. Recall that according to the SCTMA, the depression in $T_c$ should not depend on the strength of the scatterer nor on the value of the impurity phase shift. In Figure 8, we show resistivity data for a LSCO $x = 0.26$ single crystal with a $T_c$ value of 5 K, commensurate with its doping level. The residual resistivity of this sample is $\rho_0 = 50\,\mu\Omega$cm, giving a corresponding $\Gamma_N(0) \approx 160$ K. The latter appears far too high to sustain superconductivity within the dirty $d$-wave scenario.

One might argue, of course, that the doping level is actually shifted to lower doping, e.g. due to oxygen vacancies, leading to a sample that, were it cleaner, would have a much higher $T_c$. This argument does not hold, however, when one examines the $T$-dependence of $\rho_{ab}(T)$. As shown previously [102], $\rho_{ab}(T)$ below about 150 K can be approximated by the expression $\rho_0 + \alpha_1(0)T + \alpha_2 T^2$ across the entire strange metal regime of OD cuprates. Approaching room temperature, $\rho_{ab}(T)$ becomes $T$-linear again, but with a different high-temperature slope $\alpha_1(\infty)T$ [102]. While $\alpha_2$ and $\alpha_1(\infty)$ are essentially doping-independent, $\alpha_1(0)$ is found to grow linearly from zero at $p$ 0.31 to a maximum at $p^* = 0.19$ [12]. This trend is found in all families of OD cuprates studied to date, including LSCO [12], Tl2201 [13] and Bi2201 [14]. Thus, the magnitude of $\alpha_1(0)$ (more robustly, the ratio $\alpha_1(0)/\alpha_1(\infty)$ which removes any geometrical uncertainty as well as differences in the unit cell volume or FS topology between the different families) can provide a good gauge of the doping level of a particular sample [102]. For the sample shown in Fig. 8A, $\alpha_1(0)/\alpha_1(\infty) = 0.4$, consistent with a doping of $p = 0.26$. Hence, the above argument does not hold.

Fig. 8B shows the resistivity curves of a LSCO single crystal [12] and a thin film [91] with nominally the same doping level ($x = 0.23$) and similar $T_c$ values (19 K and 20 K, as determined by the mid-point of their resistive transitions). Note that the resistivity curve of the thin film has been scaled by a factor of 0.5 in order to normalize the slopes of the two curves. Even after scaling, however, the residual resistivity of the film ($\rho_0 = 50\,\mu\Omega$cm) is still 2.5 times larger than for the single crystal ($\rho_0 = 20\,\mu\Omega$cm). Nevertheless, its $T_c$ value is almost identical. Moreover, the form of $\rho_{ab}(T)$ in both samples is the same, as shown by plotting the derivative in Fig. 8C, confirming that their doping levels are essentially equivalent.

According to Lee-Hone *et al.*, for $\Gamma_N(0) \approx 50$ K, the $T_c$ of LSCO23 would be reduced from 75 K to 25 K. Hence, a shift in $\Gamma_N(0)$ from 65 K to 160 K would, according to the theory, effectively kill superconductivity outright. Yet the $T_c$ is not only the same in both samples, it is also consistent with the usual $T_c$ parabola [93]. This observation is inconsistent with

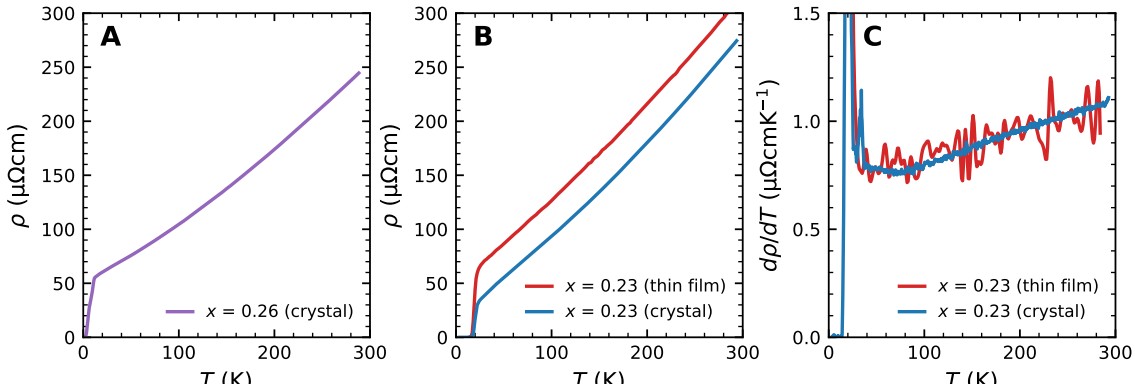

Figure 8: **A)**: In-plane resistivity of LSCO26 single crystal ($T_c \approx 5$ K). Note that the magnitude of the residual resistivity $\rho_0 = 50$ $\mu\Omega$cm. **B)**: Comparison of the in-plane resistivity of LSCO23 single crystal (blue curve) from Ref. [12] and a LSCO23 thin film (red curve) from Ref. [91]. The $\rho_{ab}(T)$ data for the thin film has been divided by 2 in order to normalize the slopes. The corresponding $\rho_0$ values are 20 and 50 $\mu\Omega$cm respectively. **C)**: Temperature derivatives $d\rho_{ab}/dT$ of the same (normalized) resistivity curves. The derivatives are identical within the scatter over the entire temperature range.

expectations from the dirty $d$-wave scenario and thus raises an important challenge to the applicability of the SCTMA treatment of BCS theory to OD cuprates.

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
