# Peer review of "Possible superconductivity from incoherent carriers in overdoped cuprates"

_SciPost Physics, doi:SciPost Phys. 11, 012 (2021)_

## Round 1 · Referee Report · Anonymous (Referee 1) · 2021-4-8

Strengths

An original idea in a pretty mature field.

Weaknesses

i) Ambiguity of concepts
ii) Incomplete presentation of data
ii) Restricted to the cuprate "bubble"

Report

The paper argues that cuprate superconductivity is an instability of incoherent carriers. The origin of high-temperature superconductivity in cuprates is a mystery several decades old. I think that the authors make several relevant observations and find a very intriguing link between the evolution of superfluid density and a subset of carriers. However, I cannot recommend the acceptance of the paper in its current form. My objections belong to three categories: i) Imprecision of language; ii) absence of connection to other strange metals or superconductors other than cuprates; and Iii) the logical flow.

i) What is “an incoherent carrier”? The authors do not define what they mean by this expression but seem to suggest that it refers to entities displaying Planckian dissipation. What about electrons in copper, which copper display a T-linear resistivity with a Planckian prefactor (see ref. 68)?

ii) Berg et al. in PNAS 117, 2852 (2020) invoke incoherent carriers in a well-defined fashion. These are “hot” carriers, which are almost classical due to their lower degeneracy temperature or shorter lifetime. Interestingly, the strange metal Sr3Ru2O7, like overdoped cuprates displays T-linear resistivity with a Planckian prefactor (ref. 68) despite its well-defined Fermi surface.

iii) Strontium titanate has also a superconducting dome. Like cuprates, its superfluid density as a function of doping decreases as soon as the peak Tc is attained (Collignon et al., PRB, 96, 224506 (2017)). This appears to be a generic feature of any superconducting dome. If for whatever reason, adding an electron, instead of enhancing Tc (because of enhanced density of states) pulls it down, then you expect the superfluid density to do the same, because adding an electron reduces foremost the ratio of the superconducting gap to the carrier lifetime. Why should one exempt cuprates from this general rule?

iv) The very first sentence of the abstract qualifies the non-superconducting state of overdoped cuprates as a strange metal. Doesn’t this contradict previous works by Hussey and collaborators, such as PRB 68, 100502 (2003)? I am also surprised by the logical flow. The paper begins with a conjecture and presents arguments in favor of this conjecture, instead of starting with observations and ending with a conclusion.

v) Fig. 1A compares two normalized quantities and finds a striking correlation. However, it raises numerous questions. What is the amplitude of superfluid density in physical units (cm-3 for example)? This information cannot be found in ref. 7. So can one compare it with the Hall density of the normal state? What is the amplitude of the residual zero-temperature specific heat and how does it evolve with doping? Was it measured on single crystals? At what temperature? Inside the superconducting state? How can rule out that this zero-temperature residual term in not caused by uncontrolled disorder? Given that the source of this information is unpublished (ref. 31) and the importance of this figure in the authors’ scenario, I recommend a more detailed presentation of the data condensed in this figure.

Requested changes

See above

  • validity: good
  • significance: high
  • originality: high
  • clarity: good
  • formatting: excellent
  • grammar: perfect

Author:  Nigel Hussey  on 2021-04-23  [id 1378]

(in reply to Report 1 on 2021-04-08)

Dear Editor,

We thank the Referee for their report, for recognizing the originality of our idea and for highlighting some areas of ambiguity and incompleteness that we address in this rebuttal and in the revised manuscript. The Referee also argued for a stronger connection to other strange metals or superconductors. We had chosen to restrict the current paper to just those two cuprate families (Tl2201 and LSCO) (a) which are single-layer, single-band cuprates, (b) whose doping range covers the entire strange metal regime and (c) for which sufficient data on the superfluid and Hall densities exist. The single-band nature of the cuprates makes the presented analysis much more straightforward than for multiband superconductors such as the iron pnictides or chalcogenides Moreover, given the unique character of the cuprates, it is unclear whether the scenario presented here would be applicable to other strange metals. Hence, we do not feel that a stronger connection to other strange metals or superconductors would necessarily strengthen the manuscript and have thus maintained the focus of the manuscript on the two highlighted cuprate families.

In the following, the original comments from the Referee are copied in red, our response is in black, while the action we have taken is summarized in blue. All changes to the actual manuscript and appendices have also been highlighted in blue. Finally, a summary of the corrections is provided at the end of the rebuttal.

Referee comment:
i) What is “an incoherent carrier”? The authors do not define what they mean by this expression but seem to suggest that it refers to entities displaying Planckian dissipation. What about electrons in copper, which copper display a T-linear resistivity with a Planckian prefactor (see ref. 68)?

Response:
We are grateful to the Referee for pointing out that we had not defined the term ‘incoherent carrier’ with sufficient clarity in our original manuscript. Hence, in the introduction of the revised manuscript and below, for the Referee’s convenience, we provide a more extensive discussion of this term.

The precise nature of these incoherent carriers is not known at present and there are many alternative definitions of incoherence in the literature. Hartnoll [Nat. Phys. vol. 11, 54 (15)], for example, defines (non-quasiparticle) incoherent transport as that controlled by the collective diffusion of energy and charge rather than by quasiparticle or momentum relaxation. Other researchers associate the term directly to a T-linear resistivity with a slope consistent with the Planckian dissipation limit. This notion, however, typically refers to high temperature T-linear resistivity and as the Referee correctly pointed out, the scattering rate in copper also exhibits a similar ‘Planckian’ form at high temperatures, in this case due to electron-phonon scattering. As the temperature is lowered and the phonon modes begin to freeze out in Cu, however, the inelastic scattering rate plummets. Indeed, the T-dependent scattering rate in Cu at 10 K is three orders of magnitude smaller than it is in cuprates (previously Ref. [68], now Ref. [71]) and thus bears no relation to the large T-linear coefficient seen in cuprates at low T. Moreover, there is no quadrature magnetoresistance (MR) linked to the Planckian dissipation in Cu, nor any other evidence for strange metal transport. The key point for our discussion is that the T- (and H)-linear MR in overdoped curpates persist both down to the lowest temperatures measured and over a wide doping range.

A coherent quasiparticle is one whose decay rate Γ is smaller than its excitation energy ε which, in a Fermi-liquid (FL), is guaranteed at sufficiently low temperatures (energies) by the relation Γ ~ ε^2. In strange metals, on the other hand, quasiparticle decoherence is implicit in the fact that Γ (or ρ) varies linearly with T and ε at the lowest energy scales. In this regard, it is worth mentioning the angle-resolved photoemission spectroscopy study of overdoped LSCO by Chang et al., [Nat. Commun. vol. 4, 2559 (13)]. Their study showed that true FL quasiparticles with Γ ~ ε^2 exist only around the nodal regions of the Fermi surface, while near the anti-nodal region, Γ ~ ε, signifying low-lying excitations with a vanishing quasiparticle residue. The resulting picture of anisotropic quasiparticle breakdown, including the extent of the FL-like and non-FL-like sectors on the Fermi surface, is strikingly similar to the one invoked in the present manuscript.

In addition to the above, very general definition, we also consider here a more empirical definition for incoherent carriers, namely electronic states that exhibit signatures of non-orbital, but nonetheless ‘metallic’ transport; specifically a H-linear MR at high field strengths that is insensitive to both field orientation and impurity scattering rate and a vanishing Hall conductivity. This insensitivity to field orientation and disorder is in marked contrast to expectations from conventional Drude or Boltzmann transport theory in which the magnitude of the MR is dictated both by the projection of the Lorentz force and by the product of the cyclotron frequency and scattering rate ωcτ, where τ includes both the inelastic and elastic (i.e. impurity) scattering rates. The observed H/T scaling of the MR also implies a direct link between this non-orbital MR and the T-linear resistivity, and by association, a direct link between the incoherent carriers and Planckian dissipation. In this way, the various definitions of incoherent transport (signatures of non-orbital response in MR and linearity of Γ/ρ on T and ε at low T, ε) become inter-connected. While Planckian T-linear resistivity is not, by itself, a sufficient condition for incoherence, we hope that it is now clear how it can be associated with the incoherent carriers.

Revision to manuscript:
A separate paragraph outlining of our definition of the term ‘incoherent carrier’ is now included in the introduction of the revised manuscript. A couple of sentences have also been added discussing the similarity between ARPES on LSCO by Chang et al..

Referee comment:
ii) Berg et al. in PNAS 117, 2852 (2020) invoke incoherent carriers in a well-defined fashion. These are “hot” carriers, which are almost classical due to their lower degeneracy temperature or shorter lifetime. Interestingly, the strange metal Sr3Ru2O7, like overdoped cuprates displays T-linear resistivity with a Planckian prefactor (ref. 68) despite its well-defined Fermi surface.

Response:
Inspection of the article by Mousatov, Berg and Hartnoll reveals no mention of the term ‘incoherent’ nor ‘incoherence’. Hence, one cannot use the description of ‘hot’ carriers in this paper as a precise definition of incoherent carriers. The key element of the Mousatov model is a large density of states (DOS) on one of the bands, as found near a van Hove singularity (vHs). When the Fermi level is tuned through this region of high DOS, the resistivity can remain T-linear down to the lowest temperatures. In this picture, the high DOS in the hot (h) regions generate significant scattering of the ‘cold’ (c) quasiparticles located elsewhere on the Fermi surface through what the authors termed cc-ch scattering.

In the cuprates, while the Fermi level is known to cross the vHs in LSCO near p*, thereby generating similar conditions for cc-ch scattering, the vHs in Tl2201 is located at a doping level far beyond the end of the superconducting dome (see Ref. [14] of present manuscript). Moreover, while in overdoped LSCO, the Fermi level is tuned away from the vHs with increasing hole doping, the opposite is true in Tl2201, yet the resistivity of both families evolves in a very similar way with doping. This implies that the Mousatov mechanism for generating T-linear resistivity is not applicable to overdoped cuprates.

Revision to manuscript:
No change to the manuscript has been made in response to this comment.

Referee comment:
iii) Strontium titanate has also a superconducting dome. Like cuprates, its superfluid density as a function of doping decreases as soon as the peak Tc is attained (Collignon et al., PRB, 96, 224506 (2017)). This appears to be a generic feature of any superconducting dome. If for whatever reason, adding an electron, instead of enhancing Tc (because of enhanced density of states) pulls it down, then you expect the superfluid density to do the same, because adding an electron reduces foremost the ratio of the superconducting gap to the carrier lifetime. Why should one exempt cuprates from this general rule?

Response:
We thank the Referee for bringing to our attention the interesting paper by Collignon et al. This very careful study of the variation of the superfluid density as a function of doping in SrTi1-xNbxO3 shows clearly how ns(0) decreases relative to nH(0) – the Hall carrier density – due to a crossover from the clean to the dirty limit. Such a crossover has frequently been invoked to explain the drop in ns(0) with doping in overdoped (OD) cuprates, as described in detail in appendix C. However, the evolution of ns(0) and nH(0) in SrTi1-xNbxO3 and in the cuprates has one crucial difference. While in both cases nH(0) increases with increasing doping beyond the optimal doping, in contrast to SrTi1-xNbxO3, the value of nH(0) in cuprates (Ref. [14]), is significantly smaller than the total carrier density expected from the full Luttinger count. It is this feature of the Hall response in OD cuprates that suggests the presence of two distinct charge sectors; something that has not, to our knowledge, been reported in SrTiO3. The different scenarios for SrTiO3 and the cuprates likely stem from the fact that the parent state is a band insulator in the former and a Mott insulator in the latter. One of the key motivations of the present manuscript is to present an alternative scenario to the dirty-limit BCS case that, for the reasons highlighted above, may be inapplicable to the OD cuprates. The seminal work by Božović et al. (Ref. [7]) also raises a serious challenge to the appropriateness of the dirty-limit picture for OD cuprates, as does the quantitative comparison of the effects of disorder (via the residual resistivity) on Tc, again highlighted in appendix C.

In light of the Referee’s comment, an extra discussion section has been added highlighting the difference between SrTiO3 and the cuprates, along with a citation to the paper by Collignon et al.

Revision to manuscript:
Extra sentences added at the bottom of page 5 highlighting the difference between SrTiO3 and the cuprates, along with a reference to the paper by Collignon et al..

Referee comment:
iv) The very first sentence of the abstract qualifies the non-superconducting state of overdoped cuprates as a strange metal. Doesn’t this contradict previous works by Hussey and collaborators, such as PRB 68, 100502 (2003)? I am also surprised by the logical flow. The paper begins with a conjecture and presents arguments in favor of this conjecture, instead of starting with observations and ending with a conclusion.

Response:
We apologise for the confusion in the original wording of our opening sentence ‘The non-superconducting state of overdoped cuprates is conjectured to be a strange metal comprising two distinct charge sectors’. Firstly, we are not referring to the state beyond the end of the superconducting dome (i.e. a non-superconducting cuprate, which does indeed show correlated but conventional Fermi-liquid behaviour). Rather, we are referring to the normal state of overdoped but nonetheless superconducting cuprates, i.e. the state above Tc or above Hc2. Hence, there is no contradiction with the previous work reported in PRB 68, 100502 (2003), which focused on a doping level beyond the superconducting dome. Secondly, in our opening statement, we used the word ‘conjecture’ inappropriately. There is now compelling evidence for the existence of two distinct contributions to the in-plane transport in overdoped cuprates, firstly from our high-field study of the Hall number [Nat. Phys. (2021) doi:10.1038/s41567-021-01197-0] and secondly from our more recent magnetoresistance scaling study [Nature, in press (2021) [2012.01208] (arxiv.org)].

As stated clearly in the Conclusion section, we took these observations as our initial motivation and then set out to address the question: which sector or sectors form the superconducting condensate? Hence, we do not believe there is any issue with the logical flow of our paper. However, in response to the Referee’s welcome comment, we have reworded the opening sentence of the abstract.

Revision to manuscript:
Opening sentence of the abstract changed to: ‘There is now compelling evidence that the normal state of superconducting overdoped cuprates is a strange metal comprising two distinct charge sectors,…’

Referee comment:
v) Fig. 1A compares two normalized quantities and finds a striking correlation. However, it raises numerous questions. What is the amplitude of superfluid density in physical units (cm-3 for example)? This information cannot be found in ref. 7. So can one compare it with the Hall density of the normal state? What is the amplitude of the residual zero-temperature specific heat and how does it evolve with doping? Was it measured on single crystals? At what temperature? Inside the superconducting state? How can rule out that this zero-temperature residual term in not caused by uncontrolled disorder? Given that the source of this information is unpublished (ref. 31) and the importance of this figure in the authors’ scenario, I recommend a more detailed presentation of the data condensed in this figure.

Response:
We thank the Referee for highlighting our oversight in not providing a more detailed description of the specific heat data contained within Figure 1A. In response to this, we have added a new appendix (A7) describing in more detail the specific heat analysis plotted in Figure 1A of the main manuscript. In support of this, a new table (Table 5) has been added showing the absolute values of the residual specific heat reported by Wade in Ref. [31]. The relevant figure is attached (Wade_specific_heat_plot.pdf) for the benefit of the Referee and of the interested reader.

The Referee also asks what is the amplitude of the superfluid density in Figure 1A in physical units, but then refers to Ref. [7] of the manuscript. Figure 1A presents data on overdoped Tl2201 while Ref. [7] describes measurements of the superfluid density in overdoped LSCO. For the sake of completeness, we address this question for both systems.

In OD Tl2201, measurements of 1/λab(0)^2 – the inverse square of the in-plane penetration depth - have been reported by three separate groups and their absolute values are listed in Table 4. Eq. (2) of the main manuscript then gives the equation linking the value of 1/λab(0)^2 to the superfluid density ns(0), normalized to a single unit cell. Again, these values are listed in Table 4. In all cases, we have assumed, as explained in the main text, that the effective mass m* = 5.2 me consistent with both quantum oscillation and specific heat studies. Hence, we do not believe there is any ambiguity around how the values of ns(0)/(1+p) plotted in Figure 1A were obtained. In response to the comment about units, however, wherever ns(0), nH(0) and ncoh are introduced in the text as stand-alone quantities, we now stress that these values have been normalised to a single Cu site.

For OD LSCO, the values of 1/λab(0)^2 listed in Table 9 (now Table 10, see below)) and quoted from Ref. [37] are the as-measured values. In Ref. [7], values for the superfluid phase stiffness ρs are quoted in units of Kelvin, but these are determined directly from the as-measured 1/λab(0)^2 values and the scaling prefactor (A) is quoted in Ref. [7], making it possible to ‘re-engineer’ the 1/λab(0)^2 values directly from the ρs values plotted in Figure 2 of Ref. [7]. Conversion of 1/λab(0)^2 into ns(0) is not as straightforward in LSCO as it is in Tl2201, due to the strong curvature of the Fermi surface, proximity to the vHs and resultant strong anisotropy of the effective mass. Our method of doing this is explained at length in the manuscript.

Revision to manuscript:
New appendix (A.7) has been added describing in more detail the specific heat analysis plotted in Figure 1A of the main manuscript. In support of this, a new table (Table 5) has been added showing the absolute values of the residual specific heat reported by Wade in Ref. [31]. All subsequent tables have been re-labeled. For further clarification of our analyses of LSCO, an extra paragraph and two citations have been added to appendix B.1 to discuss the correction factor used to account for discrepancies between ARPES- and tight-binding derived Fermi surfaces. Finally, wherever ns(0), nH(0) and ncoh are introduced in the text as stand-alone quantities, we now stress that these values have been normalised to a single Cu site.

List of corrections:

In response to Referee’s comments (labelled by roman numerals):

(i) Separate paragraph on page 3 outlining of our definition of the term ‘incoherent carrier’ is now included in the introduction of the revised manuscript.

(ii) No change to the manuscript has been made in response to this comment.

(iii) Extra sentences added on pages 5 and 6 highlighting the difference between SrTiO3 and the cuprates, along with a reference to the paper by Collignon et al..
(iv) Abstract: Line 1: changed to: ‘There is now compelling evidence that the normal state of superconducting overdoped cuprates is a strange metal comprising two distinct charge sectors,…’

(v) New appendix (A.7) added describing the specific heat analysis plotted in Figure 1A of the main manuscript. In support of this, a new table (Table 5) also added showing the absolute values of the residual specific heat reported by Wade in Ref. [31]. All subsequent tables have been re-labeled.

(v) Wherever ns(0), nH(0) and ncoh are introduced in the text as stand-alone quantities, we now stress that these values have been normalised to a single Cu site.

Additional modifications to improve clarity

Page 8: A couple of sentences added discussing the similarity between ARPES on LSCO by Chang et al.. Sentence referring to this work also added to the figure caption of Figure 2.

Page 11: Extra sentences and references added in discussion of the observation of quantum oscillations in highly overdoped Tl2201 to emphasise the fact that the observation of QOs is no longer seen as a ‘smoking gun’ for the existence of a fully coherent Fermi surface.

Page 23: Extra sentences and references added to appendix B.1 to discuss the correction factor used to account for discrepancies between ARPES- and tight-binding derived Fermi surfaces. Finally, wherever ns(0), nH(0) and ncoh are introduced in the text as stand-alone quantities, we now stress that these values have been normalised to a single Cu site.

Attachment:

Wade_specific_heat_plot.pdf

Anonymous on 2021-04-26  [id 1382]

(in reply to Nigel Hussey on 2021-04-23 [id 1378])
Category:
remark

The reply contains numerous relevant and satisfactory answers. I think the paper has been much improved and I would like to recommend the publication of the revised version.
Nevertheless, I would like to add two comments:
i) There are two well-understood ways to get a T-linear resistivity. One is electron-phonon scattering above the Debye temperature in a Bloch-Gruneisen picture (like copper). The second is electron-electron scattering when one of the two colliding electrons is classical (like Sr2Ru3O7). In both, the perfector of the T-linear resistivity is of the order of k_B/hbar. The first is obviously not relevant here as argued by the author. I am less sure about the second.
ii) The attached specific heat data is extremely useful. But the figure gives the misleading impression that the constant normal-state electronic specific heat as a function of doping has been directly measured. which is most probably not the case. It would be extremely helpful to tell the reader is that what has been measured is the relative amplitude of the JUMP in specific heat normalized to the normal state assuming that the normal state specific heat remains unchanged after the subtraction of a phononic background. Adding this detail would make the figure less impressive, less surprising, and more understandable. The critical temperature dives towards zero with overdoping and so does the entropy difference between the normal and superconducting states.

---

## Round 1 · Referee Report · Anonymous (Referee 2) · 2021-5-28

Strengths

1 )Comprehensive overview of multiple probes of electronic density in a broad doping range spanning critical doping and across multiple families of cuprates.

2) The experimental review is clearly focused around a single physical insight.

Weaknesses

The physical insight is not rooted in the data presented.

Report

The manuscript presents a review of several probes of electronic density in a broad doping range. These measurements are synthesized to support the argument that the superconductivity in the cuprates emerges directly from the incoherent part (non-Fermi surface) part of electronic spectral weight.

The main argument is that neither density n_H, as inferred from Hall measurements using the Fermi liquid phenomenology, or the density n_s, as inferred from superfluid density using BCS phenomenology, correspond to the total electronic density 1+p (holes) expected for cuprates well into the overdoped regime. The Authors observe that although n_H(p) and n_s(p) differ from 1+p in the magnitude and their stronger doping dependence, the sum total of n_H(p) and n_s(p) does check with 1+p within the accuracy of the data. They then argue that the neat partition of 1+p into n_H and n _s in a broad doping range above critical doping might suggest that n_H and n_s correspond to spectral weights of two distinct but coexisting electronic excitations, n_H to quasiparticles on the Fermi surface and n_s to the excitations outside of the Fermi surface ("incoherent"), and that because n_s is more directly associated with the superconductivity -- both in it physical interpretation and in its doping dependence -- the superconductivity must emerge directly from the incoherent part of the electronic excitations.

The argument n_H + n _s = 1+p in Tl2201 is strongly dependent on the validity of the analysis of Hall resistivity in Ref. 14 where the Hall coefficient has been found to change by about a factor of two between p=0.27 and p=0.23, much larger than the relative change in 1+p in the same doping range. The magnitude of the Hall coefficient n_H in Ref 14 has been inferred form the high field measurement which show a strong field and temperature dependence of R_H, with the variance comparable with with the factor of 2 required to distinguish reliably the value of n_H(p) and 1+p at p=0.23. Although the inferred value of n_H(p=0.27 ) in Ref 14 is consistent with the quantum oscillations measurements (Ref 29) in Tl2201, no quantum oscillation measurements exist at p=0.23 and the the uncertainty of the value of n_H in Ref. 14 cannot be reliably established.

The main point argued in this manuscript does present several interesting possibilities for the understanding of the physics of cuprates, and will be met with interest by its readers. However, the argument in the Manuscripts relies on several weak interpretational steps (BCS-like interpretation of n_s, Fermi-liquid-like interpretation of Hall resistivity) to make a strong leap in their interpretation o the nature of the superconducting state. In particular, it is not clear what is the basis for interpretation of penetration depth measurements in terms of electronic density in the absence of any quantitative description of the superconductivity emerging from incoherent excitations. It is also not clear how big is the interpretational error bar on n_H in Ref. 14.

That said, the range of experimental studies collected together in this manuscript will be of broad interest and will stimulate further discussion of the physics of cuprates.

---

## Round 2 · Author Response

We thank the Referees for their expert report(s). Following our response to the first round of reviewing, Referee #1 recommended publication of the revised version which we have now submitted following the addition of some minor revisions made in response to the final round of refereeing. Our response to Referee #1’s second set of comments as well as the points raised by Referee #2 are addressed here, followed by a list of changes to the manuscript. The changes themselves are highlighted in blue in the revised manuscript.
Reply to the second set of Comments from Referee #1
Referee comment:
The reply contains numerous relevant and satisfactory answers. I think the paper has been much improved and I would like to recommend the publication of the revised version. Nevertheless, I would like to add two comments:
Response:
We thank the Referee for recommending the publication of the revised version of the manuscript, which we have attached with this second rebuttal. We address the final two comments below.
Referee comment #1.1:
There are two well-understood ways to get a T-linear resistivity. One is electron-phonon scattering above the Debye temperature in a Bloch-Gruneisen picture (like copper). The second is electron-electron scattering when one of the two colliding electrons is classical (like Sr2Ru3O7). In both, the perfector of the T-linear resistivity is of the order of k_B/hbar. The first is obviously not relevant here as argued by the author. I am less sure about the second.
Response:
We thank the Referee for raising this point. In our recent article [Culo et al., Phys. Rev. Res. vol. 3, 023069 (21)], we discussed at length the possible relevance of the ‘cold-cold-to-cold-hot’ (cc-ch) scattering model of Mousatov et al. [PNAS vol. 117, 2852 (20)] – originally proposed to explain T-linear resistivity in Sr3Ru2O7 - to the strange metal phase of the iron chalcogenide family FeSe1-xSx. The key point in the Mousatov picture is the presence of a van Hove singularity (vHs), or region of high density of states, lying just below the Fermi level E_F . Above a certain temperature (determined by the distance \epsilon_h of the vHs from E_F and its width W_h), the carriers on the small, heavy FS become nondegenerate, i.e., classical and ‘hot’. As a result, electrons on the ‘cold’ FS are more likely to be scattered into these hot spots. In this circumstance, T -linear resistivity is realized due to the nondegenerate nature of the hot electrons. Once k_BT < \epsilon_h + W_h, electrons at the hot spots also become degenerate and the usual T^2 behavior is restored.
There are two key differences between the Mousatov picture for Sr3Ru2O7 and the cuprates. Firstly, while in LSCO, the Fermi level is tuned through the vHs at a doping level around p = 0.19, in the Tl2201 system, the vHs is not expected to be crossed until a much higher doping p ~ 0.45 [see Putzke et al., Nat. Phys. AOP (21)]. Hence, as one dopes across the strange metal region (0.20 < p < 0.30), E_F in the LSCO system is tuned away from the vHs and while in Tl2201, E_F is tuned towards it (though never crossing). The evolution of the T-linear coefficient with doping in both systems, however, is very similar. Moreover, the T-linear resistivity persists down to the lowest temperatures at all doping levels studied, meaning that there is no crossover in the form of the resistivity to a T^2 dependence in either LSCO or Tl2201, as one might expect when one is tuning away from or towards the vHs. Finally, the separation of E_F from the vHs in superconducting Tl2201 is such that the density of states in Tl2201 can be modelled as essentially isotropic within the (kx, ky) plane, as indicated in Figure 1A of the main manuscript. For these reasons, we do not believe that the Mousatov picture is applicable to overdoped cuprates. Electron-phonon scattering is not relevant here either, given the persistence of the T-linear resistivity to temperatures up to three orders of magnitude smaller than the Debye temperature. Hence, the most appropriate interpretation for the T-linear resistivity and other strange metal properties is indeed the one based on maximum dissipation, presumably due to proximity to a quantum critical point or phase, and the incoherent transport associated with it. Nevertheless, we agree with the Referee that it would be helpful to discuss different possible origins of T-linear resistivity and to examine their applicability in overdoped cuprates. We have therefore added one paragraph at the top of page 3 with a discussion about different origins of T-linear resistivity and a short explanation as to why we think that incoherent transport at the maximum dissipation limit is the most plausible.
Changes made to manuscript:
New paragraph added at the top of page 3 describing the different possible origins of a Planckian T-linear resistivity, as well as several new references: Bruin et al, Science vol. 339, 804 (13), Mousatov et al, PNAS vol. 117, 2852 (20), Zaanen, Nature vol. 430, 512 (04), Hartnoll, Nat. Phys. vol. 11, 54 (14), Zaanen, SciPost Phys. vol. 6, 061 (19) and Ledbetter, Physica C vol. 235-240, 1325 (94). Several minor changes throughout the introduction were also made for clarity.
Referee comment #1.2:
The attached specific heat data is extremely useful. But the figure gives the misleading impression that the constant normal-state electronic specific heat as a function of doping has been directly measured. which is most probably not the case. It would be extremely helpful to tell the reader is that what has been measured is the relative amplitude of the JUMP in specific heat normalized to the normal state assuming that the normal state specific heat remains unchanged after the subtraction of a phononic background. Adding this detail would make the figure less impressive, less surprising, and more understandable. The critical temperature dives towards zero with overdoping and so does the entropy difference between the normal and superconducting states.
Response:
We thank the Referee for raising this important point, but according to the thesis of Matthew Wade, and to the associated published article [Wade et al., J. Supercon. vol. 7, 261 (94)], the constancy of the normal-state electronic specific heat across the entire doping range is in fact a robust finding. To quote from the thesis of Matthew Wade (here, δ refers to the oxygen off-stoichiometry, γ_n is the normal state electronic specific heat coefficient, fig. 6.7 is identical to the main panel of figure 4 in the J. Supercon. article, while fig. 6.9 is the figure uploaded in the previous round of refereeing.):
“Not only does γ_n(T) appear to be independent of temperature, but from fig. 6.7 we can see that it also appears to show no δ-dependence within the scatter. It ought to be pointed out that although Δγ (i.e. the left hand axis on fig. 6.7) is reliably known, the value of γ (i.e. right hand axis on fig. 6.7) is less certain. The fact that γ_n shows no dependence on δ however follows from Δγ rather than γ and so is reliably known.”
“The scatter seen in γ_n(T) as a function of δ may well be the result of an inadequate phonon correction arising from a small error in δ. As a result of the rather involved process used to determine δ it is difficult to estimate the incurred error but it is quite likely to be less than the small ad hoc (|δ| < 0.005) shifts which would be required in order to force γ_s - γ_n = 0 immediately above T_c in fig. 6.7. Reanalysing the raw data letting δ vary slightly to ensure entropy conservation below T_c and γ_s = γ_n above it gives the plot seen in fig. 6.9.”
In light of the Referee’s comment, however, we have added an extra sentence stressing that the doping independence of γ_n is robust to small changes in the phonon specific heat.
Changes made to manuscript:
Extra sentence added to Appendix A.7 clarifying that the doping independence of γ_n is robust to small changes in the phonon specific heat.
Reviewer #2
We thank the Referee for their thoughtful report on our manuscript and for acknowledging the potential interest and impact of our findings. The Referee raised a couple of points that we address below:
Referee comment #2.1:
The argument n_H + n _s = 1+p in Tl2201 is strongly dependent on the validity of the analysis of Hall resistivity in Ref. 14 where the Hall coefficient has been found to change by about a factor of two between p=0.27 and p=0.23, much larger than the relative change in 1+p in the same doping range. The magnitude of the Hall coefficient n_H in Ref 14 has been inferred form the high field measurement which show a strong field and temperature dependence of R_H, with the variance comparable with with the factor of 2 required to distinguish reliably the value of n_H(p) and 1+p at p=0.23. Although the inferred value of n_H(p=0.27) in Ref 14 is consistent with the quantum oscillations measurements (Ref 29) in Tl2201, no quantum oscillation measurements exist at p=0.23 and the the uncertainty of the value of n_H in Ref. 14 cannot be reliably established.
Response:
We thank the Referee for raising their concern that the uncertainty of the value of n_H in Ref. [14] cannot be reliably established. It is important to realize, however, that the field dependence of R_H in Tl2201 (in the field-induced normal state) diminishes with decreasing temperature (Fig. 1 of Ref. [14]) and for most samples, essentially vanishes at the lowest temperatures. Moreover, while the Referee emphasizes the factor of 2 change in n_H between p = 0.27 and 0.23, it should be acknowledged that this variation in n_H is part of a larger variation of a factor of 4 between p = 0.27 and p = 0.20. These changes are much larger than the changes in n_H due to field or temperature (in the relevant ranges of interest – Fig. 2 of Ref. [14]). These large variations in n_H(0) with doping in Tl2201 are found to be consistent with those determined in Bi2201, reported both in Ref. [14] and in Lizaire et al. [arXiv:2008.13692] where the field dependence is even weaker (due to a shorter mean-free-path). Significantly, in the Lizaire study, the change in R_H(0) with doping is much larger than the changes in either temperature or magnetic field strength. Hence, we are confident that the changes in n_H(0) across the strange metal regime reported in Ref. [14] are indeed greater than the changes in 1 + p. In response to the Referee’s comment, we have added a short sentence emphasizing this point.
Changes made to manuscript:
Sentence added to the discussion of the Hall number on page 3 emphasizing that the variation in n_H(0) with doping in Tl2201 and Bi2201 is much greater than the variation in field at the lowest temperatures.
Referee comment #2.2:
The main point argued in this manuscript does present several interesting possibilities for the understanding of the physics of cuprates, and will be met with interest by its readers. However, the argument in the Manuscripts relies on several weak interpretational steps (BCS-like interpretation of n_s, Fermi-liquid-like interpretation of Hall resistivity) to make a strong leap in their interpretation of the nature of the superconducting state. In particular, it is not clear what is the basis for interpretation of penetration depth measurements in terms of electronic density in the absence of any quantitative description of the superconductivity emerging from incoherent excitations. It is also not clear how big is the interpretational error bar on n_H in Ref. 14.
That said, the range of experimental studies collected together in this manuscript will be of broad interest and will stimulate further discussion of the physics of cuprates.
Response:
Again, we thank the Referee for raising this point. The main purpose of our present manuscript was not to provide a microscopic description of the superconductivity emerging from incoherent excitations, but rather to highlight the anti-correlation between n_H(0) and n_s(0) as well as the empirical relation n_s(0) + n_coh = 1 + p. We accept that certain assumptions have been made in order to arrive at these (cor)relations, but throughout the manuscript, we have strived to be fully transparent and explicit in the assumptions that we have made. It is indeed our hope and expectation that these relations will be of broad interest and will stimulate a response from the theoretical community working on the origins of superconductivity in the cuprates and in other unconventional superconductors.
Our estimates of n_s(0) were made using the London equation and thus do not require any recourse to BCS theory. Moreover, the excellent agreement between the estimates of n_s(0) and the change in the electronic specific heat coefficient below T_c suggests strongly that our approach (interpreting penetration depth measurements in terms of electronic density of states) is valid. The conversion of the Hall resistivity into a carrier density may seem at first sight difficult to justify in a system close to the Mott insulating state. However, Ando and co-workers measured the Hall response in lightly-doped LSCO and found that the low-T Hall number n_H(0) ~ x (p) for 0.01 < x (p) < 0.08 [Ando et al., PRL vol. 92, 197001 (04)]. Beyond x = 0.08, this correspondence breaks down, presumably due to the emergence of charge (stripe) order in the intermediate doping range 0.09 < x (p) < 0.16. At high doping (p > 0.27), the relation n_H(0) ~ 1 + p has been confirmed in Tl2201. Hence, at both ends of the phase diagram, the relation between n_H(0) and the number of mobile holes appears to hold and it thus seems reasonable to assume that in the crossover regime 0.16 < p < 0.27, the value of n_H(0) also provides a good estimate of the effective carrier density. The only assumptions made in the present study in determining the density of coherent carriers is that the Fermi surface can be decomposed into distinct coherent and incoherent sections and that the incoherent part of the Fermi surface has no intrinsic Hall response, as speculated in Ref. [14]. This latter assumption is based in part on the observation that the in-plane MR in overdoped cuprates is insensitive to both the level of impurity scattering and the orientation of the magnetic field with respect to the applied current. Both observations imply that the quadrature MR scaling found in overdoped cuprates does not stem from cyclotron motion (i.e. from the Lorentz force).
In response to the Referee’s comment, a new paragraph has been added in Appendix A.1 on page 14 incorporating the above discussion. We have also added a phrase to the sentence at the end of Section 2 emphasizing that the empirical relations introduced here do not rely on the exact microscopic origin of the non-Fermi-liquid, strange metal component.
Changes made to manuscript:
Paragraph added in Appendix A.1 on page 14 discussing the relation between n_H(0) and p in underdoped LSCO and in overdoped Tl2201 and its relevance to estimates of the carrier density extracted from R_H(0) measurements within the strange metal (crossover) regime.
Phrase added to sentence at the end of Section 2 that now reads ‘This simple empirical relation is our central finding, one that does not rely on knowing the exact microscopic origin of the non-FL, strange metal component’.

---

## Round 2 · List of Changes

List of corrections:
In response to Referee’s comments (labelled according to the comments above)
#1.1 New paragraph added at the top of page 3 describing the different possible origins of a Planckian T-linear resistivity, as well as several new references: Bruin et al, Science vol. 339, 804 (13), Mousatov et al, PNAS vol. 117, 2852 (20), Zaanen, Nature vol. 430, 512 (04), Hartnoll, Nat. Phys. vol. 11, 54 (14), Zaanen, SciPost Phys. vol. 6, 061 (19) and Ledbetter, Physica C vol. 235-240, 1325 (94). Several minor changes throughout the introduction were also made for clarity.
#1.2 Extra sentence added to Appendix A.7 clarifying that the doping independence of γ_n is robust to small changes in the phonon specific heat.
#2.1 Sentence added to the discussion of the Hall number on page 3 emphasizing that the variation in n_H(0) with doping in Tl2201 and Bi2201 is much greater than the variation in field at the lowest temperatures.
#2.2a Paragraph added in Appendix A.1 on page 14 discussing the relation between n_H(0) and p in underdoped LSCO and in overdoped Tl2201 and its relevance to estimates of the carrier density extracted from R_H(0) measurements within the strange metal (crossover) regime.
#2.2b Phrase added to sentence at the end of Section 2 that now reads ‘This simple empirical relation is our central finding, one that does not rely on knowing the exact microscopic origin of the non-FL, strange metal component’.
Anonymous on 2021-06-10 [id 1499]
I appreciate the authors' response.
I note that at lease in two other cuprates, it has been claimed that the normal-state electronic specific heat changes with doping (See Phys. Rev. B 103, 214506 (2021)). This raises the issue of how to subtract the phonon contribution in each case.
However, I do not want to delay the publication of this paper and its fascinating observations and recommend its publication in Scipost.

---

## Editorial Decision

published